# Spatial-proteomics reveals phospho-signaling dynamics at subcellular resolution

Ana Martinez-Val [1], Dorte B. Bekker-Jensen[1,2], Sophia Steigerwald[1,3], Claire Koenig [1], Ole Østergaard [1], Adi Mehta[4], Trung Tran[4], Krzysztof Sikorski[4], Estefanía Torres-Vega [5], Ewa Kwasniewicz[6], Sólveig Hlín Brynjólfsdóttir[7], Lisa B. Frankel[7,8,9], Rasmus Kjøbsted [10], Nicolai Krogh[11], Alicia Lundby[1,5], Simon Bekker-Jensen [6], Fridtjof Lund-Johansen [4✉] & Jesper V. Olsen [1✉]

Dynamic change in subcellular localization of signaling proteins is a general concept that eukaryotic cells evolved for eliciting a coordinated response to stimuli. Mass spectrometry-based proteomics in combination with subcellular fractionation can provide comprehensive maps of spatio-temporal regulation of protein networks in cells, but involves laborious workflows that does not cover the phospho-proteome level. Here we present a high-throughput workflow based on sequential cell fractionation to profile the global proteome and phospho-proteome dynamics across six distinct subcellular fractions. We benchmark the workflow by studying spatio-temporal EGFR phospho-signaling dynamics in vitro in HeLa cells and in vivo in mouse tissues. Finally, we investigate the spatio-temporal stress signaling, revealing cellular relocation of ribosomal proteins in response to hypertonicity and muscle contraction. Proteomics data generated in this study can be explored through https://SpatialProteoDynamics.github.io.

[1] Novo Nordisk Foundation Center for Protein Research, Proteomics Program, Faculty of Health and Medical Sciences, University of Copenhagen, Copenhagen, Denmark. [2] Evosep Systems, Odense, Denmark. [3] Max Planck Institute of Biochemistry, Department of Proteomics and Signal Transduction, Martinsried, Germany. [4] Department of Immunology, Oslo University Hospital, Rikshospitalet, Postboks 4950, Nydalen 0424 Oslo, Norway. [5] Cardiac Proteomics, Faculty of Health and Medical Sciences, University of Copenhagen, Copenhagen, Denmark. [6] Center for Healthy Aging, Department of Cellular and Molecular Medicine, University of Copenhagen, Copenhagen, Denmark. [7] Danish Cancer Society, Copenhagen, Denmark. [8] Danish Cancer Society Research Center, Copenhagen, Denmark. [9] Biotech Research and Innovation Centre, University of Copenhagen, Copenhagen, Denmark. [10] The August Krogh Section for Molecular Physiology, Department of Nutrition, Exercise and Sports, University of Copenhagen, Copenhagen, Denmark. [11] Department of Cellular and Molecular Medicine, University of Copenhagen, Copenhagen, Denmark. ✉email: fridtjol@gmail.com; jesper.olsen@cpr.ku.dk

Protein function is tightly controlled in cells through multiple mechanisms. Protein activity can be dynamically modulated, for instance, by changing translation rate[1] or by post-translational modifications, such as site-specific phosphorylation or ubiquitination[2]. Moreover, most proteins do not operate in isolation, but rather they need to interact with other proteins to elicit their functions[3]. Most of these regulatory mechanisms have been the subject of extensive research. Additionally, a protein's function can also be regulated in a spatial manner by modulating its subcellular localization. This regulatory layer, i.e., cellular compartmentalization, is especially important for faithful transmission through signal transduction pathways, where fast responses are required, such as nucleocytoplasmic shuttling of transcription factors for transcriptional control[4] or endocytic internalization of activated receptors for degradation or recycling[5]. Furthermore, protein moonlighting[6] is a well-established phenomenon by which proteins exert different functions depending on their subcellular location[7,8]. Consequently, subcellular localization of proteins has been studied extensively, mainly by using molecular biology techniques, relying on either imaging[9–11], or, most recently, on information derived from proximity-labeling experiments[12–14]. Although very sensitive and powerful, these techniques lack throughput, as they cannot provide information on protein location at a proteome-wide level. In recent years, several studies presented the potential of MS-based proteomics to explore the subcellular proteome. Among them, approaches such as LOPIT-DC[15] or SubCellBarcode[16] stand out due to their sensitivity, coverage, and resolution, mapping the location of more than 8000 proteins. However, both methods rely on isobaric tandem mass tag (TMT) labeling for quantifying subcellular protein localization, and they require extensive off-line peptide fractionation and consequently lengthy liquid chromatography coupled to tandem mass spectrometry (LC-MS/MS) analysis time to achieve the desired depth on the proteome, thus minimizing throughput. Conversely, other studies have proposed single-shot LC-MS/MS analysis and label-free quantification as an alternative to obtain faster organellar maps[17,18] at the expense of coverage, mapping the location of about 4000 proteins in ~12 h of MS analysis time. Hence, there is still a gap of knowledge in how to obtain deep subcellular proteomes without compromising MS-time to enable the application of these techniques to different experimental conditions with multiple biological replicates.

Subcellular translocation of a protein is a dynamic regulatory event, and it is therefore essential to incorporate the temporal dimension when studying protein translocation in response to stimuli. Spatio-temporal proteomics is very challenging, and only few studies have been performed, such as the spatio-temporal characterization of cytomegalovirus infection[19]. This is probably due to the complexity in applying these workflows in a high throughput manner, which substantially limits their usefulness to analyze global subcellular proteome dynamics. Except for Krahmer et al[20], none of the current approaches to study the subcellular proteome changes have been extensively employed to study the signaling layer of the phospho-proteome, which is known to control and trigger protein relocation[21]. To overcome these limitations, and provide an accessible workflow to study spatio-temporal phospho-proteome regulation, we present a workflow based on a sequential subcellular fractionation protocol that when coupled to fast chromatographic LC-MS/MS analysis using data-independent acquisition (DIA) provides rapid, sensitive, and reproducible subcellular phospho-proteome maps. Importantly, our high-throughput approach allows studying spatial-dynamics in a temporal manner in both cell lines and tissues with multiple replicates. To demonstrate the general applicability of the spatial proteomics workflow, we have applied our method in two different biological settings to study the spatio-temporal response of the proteome and phospho-proteome both in vitro and in vivo.

## Results

**High-throughput and reproducible maps of subcellular phospho-proteomes using directDIA-MS spatial proteomics.** Chemical fractionation provides an attractive alternative to more elaborate methods for the separation of intact organelles. However, reproducibility is hampered by the widespread use of a plethora of kits with undisclosed composition and the lack of meta-analysis[22]. Here, we present a streamlined pipeline for the analysis of (phospho)-proteome dynamics in distinct subcellular compartments of the cell (Fig. 1a). Six extraction buffers with different detergent, salt, and chemical composition (see Methods)[23] were used sequentially to profile six distinct subcellular fractions in cell lines and tissues at both proteome and phospho-proteome level. While detergents with different solubilizing capacity have been applied earlier for chemical subcellular fractionation, an important observation made here is that better partitioning can be achieved by also varying the ionic composition of the extraction buffers. Accordingly, cells permeabilized with a digitonin-based hypotonic buffer 1 released a large number of cytoplasmic proteins when resuspended in a detergent-free solution buffer 2 with 140 mM NaCl. Successive use of Tween-20 and dodecyl maltoside (DDM) in buffers 3 and 4, respectively, provided means to separate soluble proteins in cytoplasmic organelles from membrane proteins, while exposure of membrane-stripped nuclei to a detergent-free solution with 500 mM NaCl in buffer 5 yielded a highly pure nuclear extract. The remaining insoluble material is a mixture of nucleoli, cytoskeleton, and poorly soluble membrane proteins that requires a 0.3% SDS-containing buffer 6 for solubilization. To validate the sequential extraction and integrity of various subcellular compartments at the different steps of the fractionation protocol, we performed transmission electron microscopy (TEM) imaging of each resulting fraction (Fig. 1b).

In contrast to already published MS-based subcellular fractionation methods, our approach is optimized for high-throughput analysis. Due to the use of directDIA and fast chromatographic gradients, quantitative data for subcellular proteomes and phospho-proteomes were generated for six subcellular fractions in just 5 h of MS time. This enables the possibility to include multiple biological replicates as well as different experimental conditions or time points.

To evaluate the coverage and reproducibility of the entire workflow, we applied our method to HeLa human cervix carcinoma cells. Four biological replicates of HeLa cells were serum starved overnight, and trypsinized just prior to subcellular fractionation. The entire experiment was performed at 4 °C in the presence of protease and phosphatase inhibitors. Once the fractions were collected, they were lysed in boiling SDS and reduced/alkylated before being subjected to protein aggregation capture (PAC)-based tryptic digestion[24] using the fully automated KingFisher platform. Approximately five percent of the resulting peptide mixtures from each subcellular fraction were used for total proteome analysis. Remaining peptides were subjected to phosphopeptide enrichment using TiIMAC-HP magnetic beads, also on the KingFisher platform. To increase measurement depth of single-shot subcellular proteomes, we took advantage of high field asymmetric waveform ion mobility spectrometry (FAIMS) in combination with fast scanning DIA acquisition methods[25]. Conversely, we chose more sensitive MS-acquisition settings for phospho-proteomics samples, without FAIMS and increased the MS/MS injection times and resolution (86 ms and 45,000, respectively) to maximize the detection of low

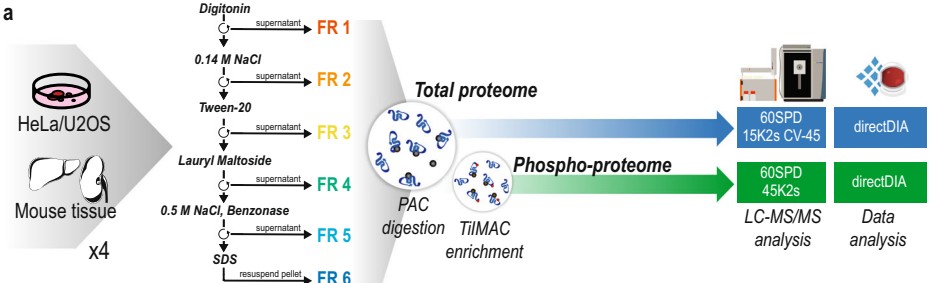

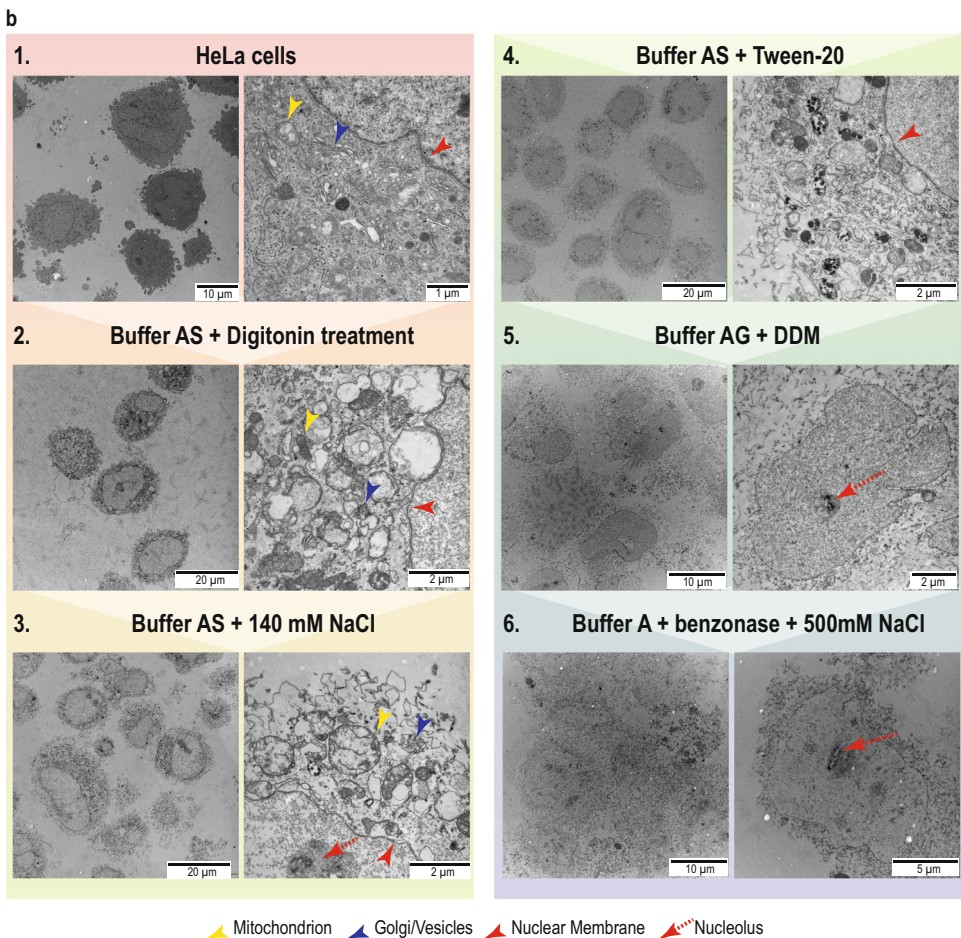

**Fig. 1 High-throughput and reproducible subcellular fractionation. a** Experimental workflow for subcellular fractionation and LC-MS/MS data acquisition. SPD: Samples Per Day, CV Compensation Voltage. **b** Transmission Electron Microscopy images from the different fractionation steps. 1: HeLa cells after trypsinization from cell culture plates. 2: HeLa cells after incubation with buffer 1 containing digitonin. 3: pellet obtained from previous step after incubation with buffer 2 containing 140 mM of NaCl. 4: pellet obtained from previous step after incubation with buffer 3 containing 0.05% Tween-20. 5: pellet obtained from previous step after incubation with buffer 4 containing 1% DDM. 6: pellet obtained from previous step after incubation with buffer 5 containing 500 mM NaCl and benzonase. Yellow arrows point to mitochondria, blue arrows point to Golgi apparatus, red arrows point to nuclear membranes and dashed red lines indicate nucleoli. Six HeLa samples were prepared in parallel for TEM acquisition, one for each subcellular fractionation step and posterior TEM imaging.

abundant species (Fig. 1a). In both cases, we used a spectral library-free approach (directDIA) in Spectronaut to analyze the resulting raw MS data. In total, we could quantify 6952 proteins and 7957 phosphorylation-sites in our dataset (Supplementary Data 1–2). On average, ~4000 proteins were identified in each fraction and more than 90% of them were robustly quantified in three out of four replicates (Fig. 2a). Importantly, 82% were reproducibly found in two or more fractions and 23% of the proteins quantified in the dataset were confidently identified in all fractions. This is in agreement with previous reports that claim

that a significant part of the proteome is not restricted to one location[9]. However, importantly, protein location cannot be directly extrapolated by merely identification, but instead from relative enrichment across fractions. In fact, our quantitative data also suggest that although proteins can be widely distributed within the cell, most of them have a predominant location or subcellular niche (Supplementary Note 1). Consequently, each fraction obtained with the current protocol captures a unique part of the cellular proteome (Fig. 2c). In contrast, the phospho-proteome profile of each fraction yielded more variable results per

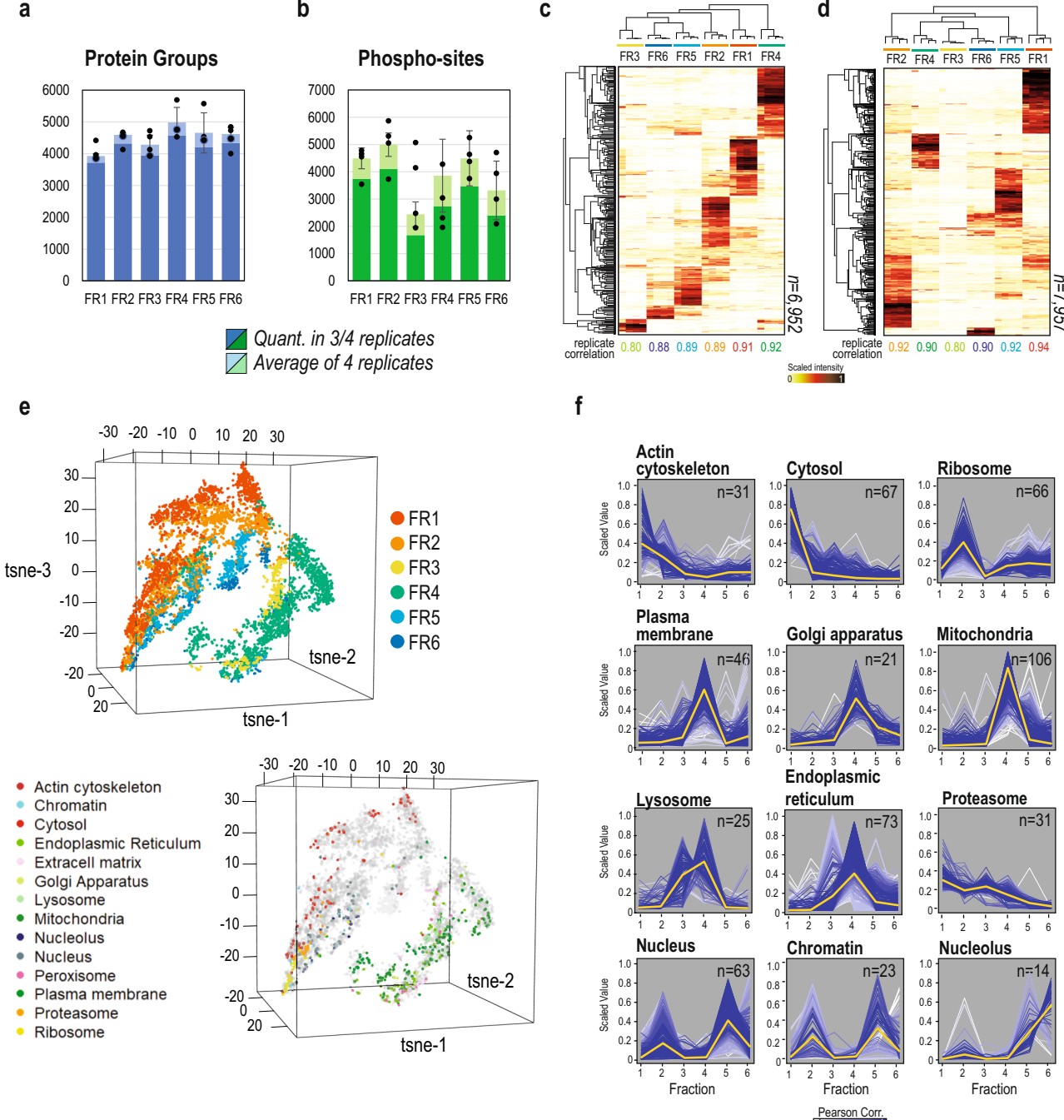

**Fig. 2 Overview of subcellular fractionation resolution and subcellular compartment enrichment. a** Bar-plot summary of the identified proteins in HeLa samples as average of 4 replicates per fraction (light blue bar) and quantified proteins in at least 3 replicates (dark blue). Height of the bars represents the mean number of identifications of $n = 4$ experimental replicates, and error bars represent the standard deviation in identification numbers between replicates. **b** Bar-plot summary of the identified phosphorylation sites in HeLa samples as average of 4 replicates per fraction (light green) and quantified phosphorylation sites in at least 3 replicates (dark green). Height of the bars represents the mean number of identifications of $n = 4$ experimental replicates, and error bars represent the standard deviation in identification numbers between replicates. **c** Heatmap of scaled intensities per replicate, of four replicates, of the subcellular proteome of HeLa, showing both protein and sample clustering. **d** Heatmap of scaled intensities per replicate, of four replicates, of the subcellular phospho-proteome of HeLa, showing both phospho-site and sample clustering. **e** t-SNE plots of log2 transformed intensities of proteins identified in the subcellular fractionation of HeLa ($n = 4$). TOP: The clusters assigned for each fraction in Fig. 2c are used to color-code it. BOTTOM: Markers of different organelles are highlighted in the t-SNE plot. **f** Profile-plots of cell compartment markers in the subcellular proteome HeLa dataset. Scaled intensity across fractions is plotted for each independent replicate. Gradient of white to blue indicates Pearson correlation to the centroid of each distribution, which is highlighted as a yellow line. "N" specifies the number of protein markers used to make the profile plots, but the plots show that number of markers for each one of the 4 replicates used experimentally. Source Data for Figs. 2a and b are provided as a Source Data file.

fraction (Fig. 2b), where fractions 3 and 6 reproducibly result in less identified phosphorylation sites, which can be due to intrinsic lack of phosphorylation events in those specific subcellular compartments.

Hierarchical clustering analysis of the scaled fractional intensities of proteins and phosphorylation sites reveals high reproducibility between experimental replicates with Pearson correlation between replicates in the range 0.80–0.94 (Fig. 2c–d, Supplementary Fig. 1a–b). Most importantly, the intensity profiles for both proteins and phosphorylation sites across fractions reveal very well-defined clusters corresponding to the proteins enriched in each cellular compartment purified in each fraction, reflecting the resolving power of the fractionation method (Fig. 2c–d). To define which cell compartment was enriched in each fraction, we annotated the dataset using established subcellular protein markers[26,27] and plotted their profile distribution across fractions at the proteome level (Fig. 2e–f and Supplementary Data 1). This analysis revealed three major cellular compartments in our dataset: fractions 1–2 corresponding to cytosolic and cytoskeletal proteins, fractions 3–4 to plasma membrane and membranous organelles, and fractions 5–6 to the nucleus (Fig. 2f). Moreover, we also observed a very clear distinction of nuclear components between fractions 5 and 6, with nucleoplasm and chromatin-bound proteins mainly purified in fraction 5, whereas nucleolar proteins are mostly observed in fraction 6 (Fig. 2f). Gene Ontology (GO) enrichment analysis in each protein cluster revealed more specific patterns for each fraction (Supplementary Fig. 1c). Interestingly, we observed that many protein complexes tend to be purified in fraction 2, including ribosomes, the exocyst or the septin complex (Supplementary Data 1). On the other hand, many proteins involved in cell cycle and DNA replication (Supplementary Fig. 1d and Supplementary Data 1) are co-purified in fraction 2, possibly due to cells undergoing cell division, in which the nuclear membrane is dissolved (Supplementary Data 1). To double-check the validity of the protein markers used to assess different subcellular localization of the proteins identified, we overlapped the dataset with the subcellular annotations from the more comprehensive imaging-based Cell Atlas[9] and found that the subcellular patterns were reproduced (Supplementary Fig. 1e). Interestingly, we did not find any specific subcellular compartment or organelle, clearly purified in fraction 3. Gene Ontology (GO) enrichment analysis of the proteins enriched in fraction 3 revealed that they belong to the lumen of several organelles including endoplasmic reticulum (q-val: 1.06e-59), lysosomal lumen (q-val: 1.38e-17) or mitochondrial intermembrane space (q-val: 4.9e-9). In fact, we observed that protein markers for lysosomes and endoplasmic reticulum (Fig. 2f) showed a secondary peak in fraction 3 corresponding to luminal proteins from both organelles (Supplementary Fig. 2a–b). As hypothesized, the use of the milder surfactant Tween-20 in buffer 3 releases soluble protein from the inner part of membranous organelles, whilst membrane-bound proteins are released in the subsequent fraction 4, after treatment with buffer containing DDM.

Next, we assessed the reproducibility of the subcellular protein localization between different cell lines by repeating the analysis in U2OS human osteosarcoma cells (Supplementary Data 3), and found the fractionation resolution to be highly reproducible indicating conserved subcellular proteome distribution across the two cell lines (Supplementary Fig. 3a). Profiling the known subcellular protein markers in U2OS cells, and comparing the correlation of their profiles against those obtained for HeLa (Supplementary Fig. 3b–c), also showed good reproducibility of the technique independently of the two cell lines.

Since the current protocol is based on the sequential fractionation of cellular compartments, we hypothesized that the integrated MS signal intensity for a given protein measured across fractions should represent its relative abundance in the whole-cell proteome. To evaluate if this is true, we plotted the sum of each protein's intensity across fractions against the intensity observed in a total lysate, and found a good correlation (Pearson correlation >0.7, p value: 4e-16) between both datasets, confirming our expectations (Supplementary Fig. 3d). However, this correlation is not as high as the one often observed between biological replicates of whole-cell lysates. This could be due to protein losses in the washes performed to minimize carryover between consecutive fractions. To assess how much protein was lost during the different washing steps, we analyzed all fractions and washes obtained with this protocol by SDS-PAGE electrophoresis. Although some protein signal is clearly observed in the washes, it is negligible in comparison with the protein signal in each of the main fractions (Supplementary Fig. 4a). To further validate the fractionation resolution between subcellular fractions, we excised the most intense gel bands observed in the SDS-PAGE gel and analyzed them by LC-MS/MS. The most abundant proteins in each gel band corresponded to markers of the compartments previously defined (Supplementary Fig. 4b). Moreover, we quantified the abundance of these marker proteins in subsequent washes and found that the abundance in the washes compared to the main fraction is very small making up few percent of total intensity at best. Therefore, we conclude that the protein losses occurring during washes can be ignored in further analysis (Supplementary Fig. 4b).

Reassuringly, the observed subcellular protein profiles were highly reproduced at the phospho-proteome level (Supplementary Fig. 5a, Supplementary Data 2), with few interesting differences. For instance, whilst ribosomal proteins are highly enriched in fraction 2 at the proteome level, their phosphorylated counterparts, such as the tri-phosphorylated S235/S236/S240 form of RPS6, are found specifically enriched in fraction 1. This observation could indicate an intrinsic difference in compartmentalization of ribosomal proteins depending on their phosphorylation status, but this needs further research.

Next, we investigated the relationship between subcellular localization and global phosphorylation events. To do so, we plotted the profiles of 221 kinases identified in our dataset (Supplementary Fig. 5b). Interestingly, we found that fraction 3 contained less kinases compared to the other fractions, which aligns well with our previous observation of lower phosphorylation events reported in this specific compartment (Fig. 2b). Furthermore, to confirm that both a kinase and its substrate were localized in the same cellular compartment, we evaluated some well-known kinase-substrate relationships and found that they generally co-localize (Supplementary Fig. 5c). To investigate the spatial relationship between kinases and their substrates, we explored the kinase activity landscape revealed by our dataset. From PhosphoSitePlus[28], we retrieved the information of known kinases for the phosphorylated sites observed in our dataset. We visualized the resulting network of kinases, grouped by the compartment in which they are most abundant, together with their substrates and their main location as well (Supplementary Fig. 5d). As expected, this analysis reveals that prominent protein kinases such as EGFR or PRKAA1 are found in the same subcellular location as their substrates (Supplementary Fig. 5e). Interesting exceptions from this are kinases like GSK3B, MAPKAPK2 or AURKA, which show more widespread activity targeting substrates from several compartments indicating multiple locations of these kinases or movement of phosphorylated substrates across compartments (Supplementary Fig. 5e).

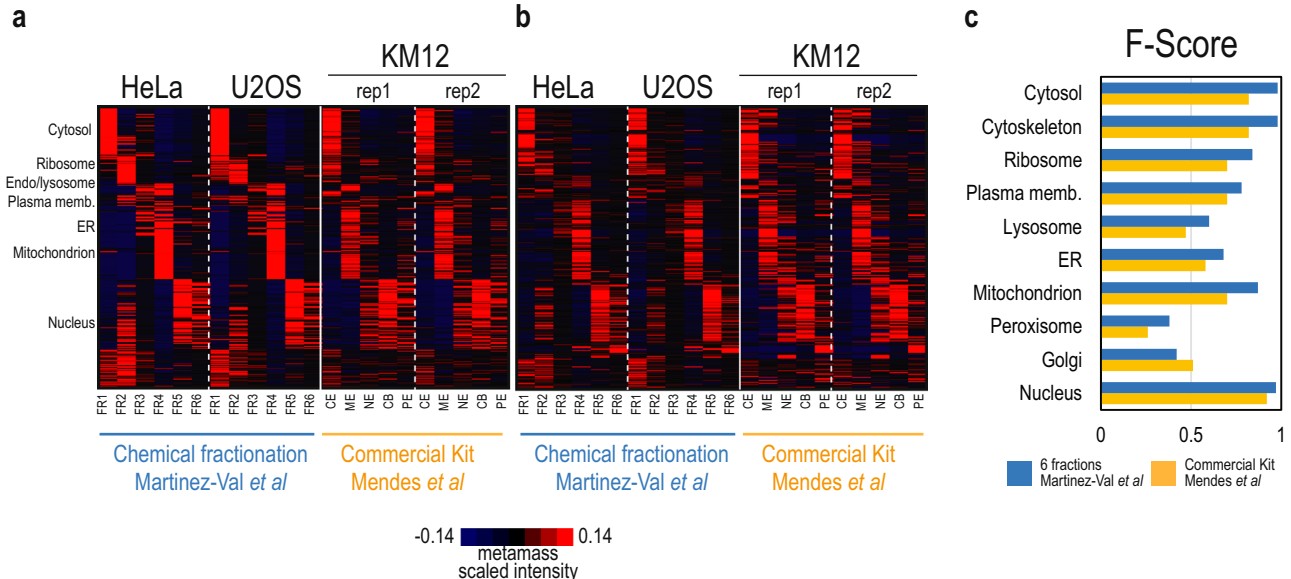

**Fig. 3 Comparative analysis of subcellular fractionation protocols based on chemical reagents using Metamass. a–b** Heatmaps showing protein distribution across fractions obtained from HeLa and U2OS using the present subcellular fractionation protocol and KM12 using the commercial kit (Pierce) employed in Mendes et al.[29]. Proteins were classified and sorted using the Excel-based analysis tool MetaMass (Supplementary Data 4). In (**a**) proteins are clustered based on the data from HeLa and U2OS fractionation in the present study. In (**b**) proteins are clustered based on Mendes et al data. Heatmaps were obtained after normalizing gene distribution and center samples by mean in Cluster 3.0, and plotted in TreeView. CE Cytoplasmic Extract, ME Membrane Extract, NE Nuclear Extract, CB Chromatin-Bound, PE Pellet Extract, ER Endoplasmic Reticulum. **c** F-score barplot for the protein assignment to organelles in the present study (blue) and in Mendes et al (yellow).

To benchmark our results against previously published datasets, we used the MetaMass[22] tool for meta-analysis of subcellular proteomics data (see Methods and Supplementary Data 4)[22]. MetaMass analyses a list of gene names and assigned groups obtained by k-means clustering of normalized MS data and compares this to several built-in sets of protein markers for subcellular compartments. The output includes statistics for precision and recall for the markers as well as the harmonic mean of the two (F-score). The markers used here correspond to subcellular locations mapped for U2OS cells by MS analysis of fractions obtained by organelle separation[9]. We compared our dataset against a reference published dataset obtained by chemical fractionation of KM12 colorectal carcinoma cells using a widely used commercial fractionation kit[29] (Fig. 3a–b). The method presented here achieved higher F-scores for partitioning of the identified marker proteins from HeLa cervical carcinoma cells and U2OS osteosarcoma cells into the correct subcellular compartments than those calculated for the reference dataset (Fig. 3c). An unexpected finding was that the F-scores for markers of the plasma membrane, endoplasmic reticulum, lysosome, and mitochondria were in the range of 0.69–0.85 (Fig. 3c). Similarly, our method reported well-defined fractions and comparable results as other published methods, such as differential density centrifugation[9,15,30] or differential centrifugation at both proteome[15,16] (Supplementary Fig. 6a–e) and phospho-proteome level (Supplementary Fig. 6f), whilst minimizing input requirements, simplifying sample preparation, and reducing MS acquisition time (Supplementary Table 1). Importantly, the precision and resolution obtained with centrifugation-based methods for organelle separation is higher than the current methodology, which, on the other hand benefits from higher throughput.

**Spatio-temporal phosphoproteomics shows in vitro and in vivo vesicle-mediated signaling in response to EGF stimulation.** The most significant advantage of our subcellular fractionation

protocol is that it enables rapid and extensive proteome and phospho-proteome measurements, which provides an excellent basis for exploring the dynamic behavior of signaling proteins in response to specific stimuli in a spatio-temporal manner. Dynamic cell signaling via Receptor Tyrosine Kinases (RTKs) represents an excellent model to study spatio-temporal protein regulation[31,32]. Epidermal growth factor receptor (EGFR) activation by ligand binding, auto-phosphorylation, recruitment of signaling adaptors, and subsequent internalization into endocytic vesicles for either degradation or recycling is a great example of how spatiotemporal dynamics in cells are controlled by rapid phosphorylation events[5]. To study EGFR signaling dynamics at the subcellular level, we stimulated HeLa cells with EGF and measured the change in the subcellular proteome and phospho-proteome at five different time points (i.e., 0, 2, 8, 20, and 90 min upon EGF stimulation). Importantly, we performed all experimental conditions in biological quadruplicates. From the resulting 240 DIA runs, we were able to confidently quantify (in at least three out four replicates) 7142 unique proteins covering different protein-coding genes (PCGs) and 11,046 phosphorylation-sites (Fig. 4a and Supplementary Data 5).

To study translocation events in response to EGFR activation, we devised a statistical method for unbiased analysis of protein translocation from quantitative spatial proteomics data termed "Movement Score" by which we calculate the magnitude of all protein translocations between fractions against the statistical significance of the changes. Applying stringent thresholds on both mobility score of at least 10% and adjusted p-value < 0.05, we can differentiate between insignificant trends and significant changes in protein translocation (Fig. 4b). From this analysis, we found GRB2, SHC1, and CBL among proteins changing locations from the cytosol to the membrane-bound lysosomes in a significant manner (Fig. 4b) at 2 and 8 min of EGFR activation, as well as novel candidates, such as PRELID1, CHUK, RNF220 or ELL2 (Supplementary Fig. 7a). This confirms established literature on EGF stimulation by which EGFR is rapidly auto-phosphorylated

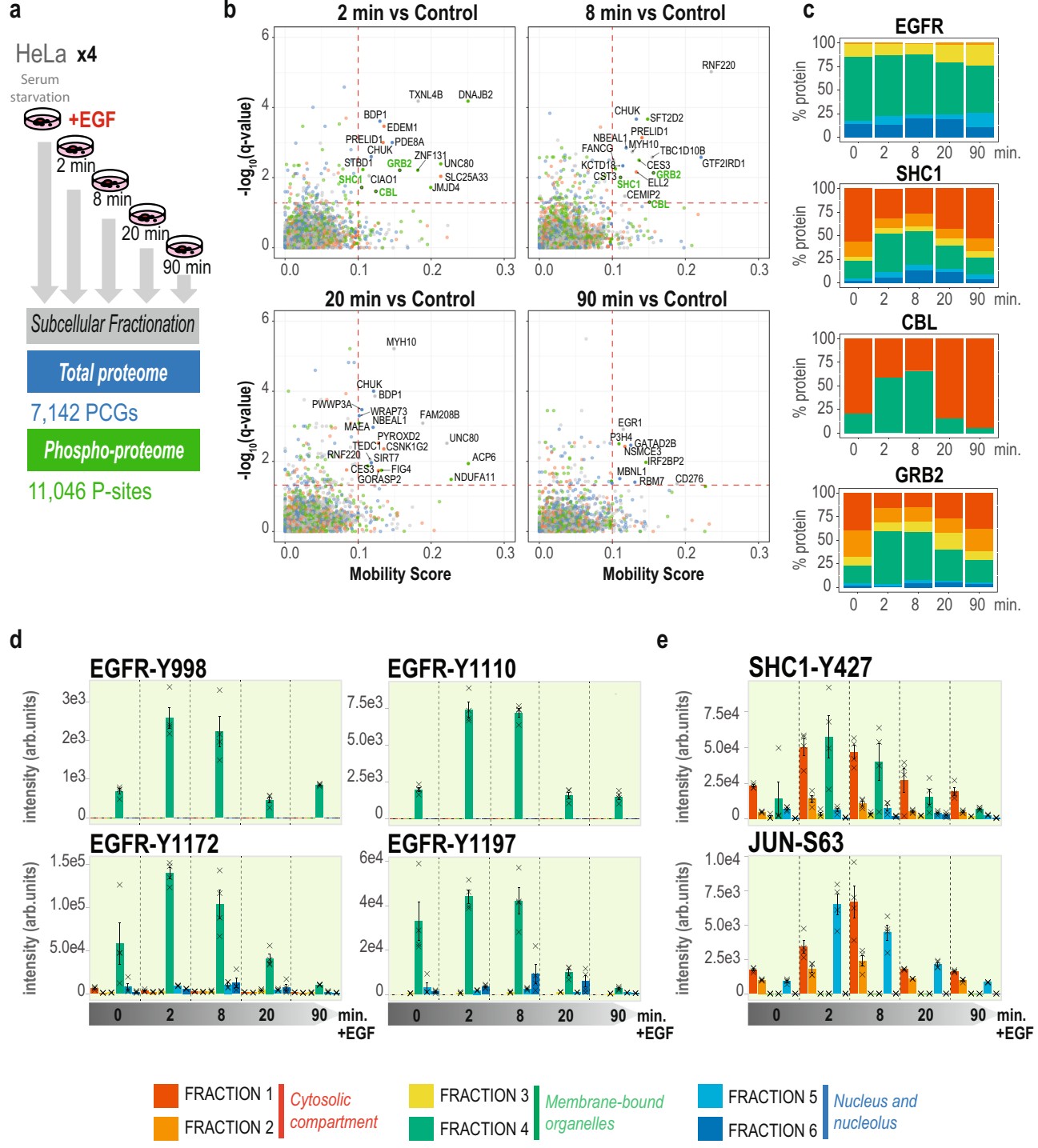

**Fig. 4 Spatio-temporal phosphoproteomics in response to EGF stimulation. a** Experimental design and result overview with number of proteins and phosphorylation-sites (p-sites) obtained from the subcellular fractionation of HeLa cells treated with EGF at different time points. PCGs: Protein Coding Genes. **b** Translocation plots of EGF treated samples at 2, 8, 20, and 90 min versus control. Red dashed lines indicate the cutoff threshold used for mobility score and significance levels. The color of the dots indicate which compartments the proteins are moving to and from. Compartment are grouped as three: cytosol (FR1 and FR2), membrane-bound organelles (FR3 and FR4) and nuclear compartment (FR5 and FR6). Red: nuclear compartment to/from membrane-bound organelles. Green: cytosol to/from membrane-bound organelles. Blue: nuclear compartment to/from cytosol. Grey: proteins moving within same compartment. GRB2, SHC1 and CBL are highlighted in green. **c** Stacked bar-plot of scaled protein intensities across fractions and time points of EGFR and adaptor proteins CBL, SHC1, and GRB2. **d** Bar-plot of intensities across fractions and time points of EGFR tyrosine phosphorylation sites. A cross represents each independent measurement. Height of the bars represents the mean intensity of $n = 4$ measurements of the protein, and error bars represent the standard error of the mean. **e** Bar-plot of intensities phosphorylation sites across fractions and time points of SHC1 and JUN. A cross represents each independent measurement. Height of the bars represents the mean intensity of $n = 4$ measurements of the protein, and error bars represent the standard error of the mean. Source Data for Figs. 4c, d, and e are provided as a Source Data file.

on its cytoplasmic tyrosine residues triggering the recruitment of adaptor proteins such as GRB2[33], SHC1[34], and CBL[35], transducers of downstream signaling. This series of events unleash the rapid internalization of the receptor into endosomes for further degradation[5], which was already observed by Itzhak and colleagues using differential centrifugation-based subcellular proteomics[18]. In our data, EGFR was mainly purified in the plasma membrane fraction (fraction 4), although we also detected it in nuclear fraction 6, which has been previously described in literature[36] (Fig. 4c). However, since our approach does not separate plasma membrane from membrane-organelles (such as endosomes) we cannot directly follow the translocation of EGFR from the plasma membrane to early endosomes. In contrast we can clearly detect how the adaptor proteins GRB2, SHC1, and CBL, which are primarily cytosolic in unstimulated cells, rapidly reduce their cytosolic presence from 2 to 8 min of EGF stimulation (Fold-change at 8 min in fraction 1: SHC1: −1.05; $q$ value = 0.024; GRB2: −1–12, $q$ value = 0.011; CBL: −2–21, $q$ value = 0.024. Q-values obtained from a moderated t-test with Benjamini-Hochberg FDR correction) (Fig. 4c and Supplementary Data 6), as they are recruited to the EGFR containing membrane fraction (Fig. 4c). Importantly, full proteome analysis of changes at those time points (2 and 8 min upon EGFR activation) do not reveal alteration of total protein abundance levels of GRB2, SHC1, or CBL, validating that the observed changes are due to translocation (Supplementary Fig. 7b). Most interestingly, we also traced the dynamic wave-like movement of these proteins across time, as they are shuttled back to the cytosol following the transient activation and degradation pattern of EGFR phospho-signaling after 20 min upon stimulation and recruitment by EGFR. This was measured directly by quantifying the dynamic phosphorylation of EGFR tyrosine residues (Fig. 4d and Supplementary Data 7), which follow the same dynamic pattern in the membrane fraction 4 as the adaptor proteins (Fig. 4c). These EGFR tyrosine phosphorylation sites act as docking sites for the SH2 domains in the adaptor proteins, which are subsequently phosphorylated, too. For example, we find that SHC1 is rapidly phosphorylated at tyrosine 427 after 2 min of EGF stimulation and that this phosphorylation is clearly observed in the membrane-bound fraction 4 (Fig. 4e). This observation indicates that SHC1 is activated upon contact with EGFR, therefore revealing that the EGFR phosphorylation of SHC1 is a direct consequence of subcellular translocation. However, a fraction of the SHC1 phosphorylation is observed to increase in the cytosolic fraction (Fig. 4e), which might suggest that known cytosolic tyrosine kinases such as SRC also triggers phosphorylation of their substrates upon EGFR stimulation[37]. Furthermore, we also detect signaling events downstream of EGFR, such as phosphorylation of the transcription factor JUN on serine 63 in its transactivation domain, which is important for its transcriptional activity showing how signaling from EGFR is transmitted into the nucleus (Fig. 4e).

Although powerful as model systems for studying cell signaling, cell lines have certain limitations for in vivo extrapolation. Therefore, to extend the scope of our spatio-temporal proteomics approach, we applied the workflow to an in vivo system. For that purpose, we performed animal experiments, where two groups of four mice were injected intravenously with saline or EGF for 10 min, respectively, followed by 1.5 min perfusion with protease and phosphatase inhibitors[38]. Whole livers and kidneys were subsequently explanted for spatio-temporal (phospho)-proteomics. The subcellular fractionation protocol was slightly modified for adaptation to organ tissues (see Methods). Briefly, tissues were homogenized in a saline buffer using the Precellys tissue homogenizer system. Following this, tissue extracts were pelleted by centrifugation, and cleaned twice

with a saline buffer before proceeding with the subcellular fractionation protocol. We were able to quantify 5677 and 6659 proteins across the six fractions in liver and kidney, respectively; as well as 5150 and 4331 phosphorylation-sites (Fig. 5a and Supplementary Data 8–9). We observed high reproducibility of the proteins purified in each fraction between tissues (Fig. 5b), and found that most of the subcellular compartments enriched per fraction reproduced what we previously observed for HeLa cells (Fig. 5d). However, we noticed few but relevant differences mainly for the fraction in which lysosomal and extracellular matrix proteins were purified (Fig. 5d and Supplementary Fig. 8a). Interestingly, we also found that mitochondrial proteins showed a completely different profile in kidney versus liver[39–41]. To explore this further, we performed transmission electron microscopy (TEM) imaging at the different stages of the subcellular protocol in both tissues. Importantly, we observed, that cell integrity is disrupted due to the homogenization step (Fig. 5c and Supplementary Fig. 8b–c), which might explain how proteins from vesicles are already observed in the first fraction. However, although the whole cell is disrupted, the homogenization preserves the structure of most organelles, such as mitochondria, Golgi apparatus and nucleus (Fig. 5c), as well as membrane-bound vesicles which remain intact until extraction with detergents is performed (Supplementary Fig. 8b–c).

To benchmark the spatio-temporal phospho-signaling observed in vivo, we extracted the spatio-temporal profiles of the cytoplasmic EGFR tyrosine phosphorylation-sites from liver phospho-proteome and compared their subcellular distribution to the corresponding data obtained from HeLa cells. As expected, we observed a significant increase in the intensity of the tyrosine phosphorylation-sites upon EGF stimulation in both liver and HeLa cells, but, interestingly, we found that in contrast to HeLa cells, these sites showed a dual distribution between cytoplasmic fraction (fraction 1) and membranous fraction (fraction 4) in liver cells (Fig. 5e). Considering the previously observed difference in the distribution of lysosomal markers between cell lines and tissues (Supplementary Fig. 8a), we wondered whether this was also the case for endosomes. We therefore examined the distribution of protein markers for early (RAB5A, EEA1), late (RAB7A), and recycling (RAB6A, RAB8A) endosomes in the liver and kidney subcellular proteomes (Fig. 5f) and, found that early endosomes markers were also co-purified in fraction 1 in the two organs examined in here (Fig. 5f). This observation can explain the difference in EGFR phosphorylation-sites observed, as activated EGFR is rapidly internalized into early endosomes, and consequently it is expected that the tyrosine phosphorylated EGFR sites would show the highest signal in the same fraction, where the early endosomes are found.

**Subcellular protein dynamics reveals ribosome accumulation upon osmotic stress.** The well-characterized EGF signaling model demonstrated the suitability of our approach to measure rapid protein translocation as a consequence of activated phospho-signaling networks. We therefore decided to employ our methodology to identify subcellular relocation events triggered in response to cellular stress signaling. Specifically, we focused on the cellular response to hypertonic stress, which has already been described as a cause of protein translocation in neuronal cells[42]. To induce hyperosmotic stress conditions, we treated U2OS cells for one hour with 500 mM sorbitol, a natural osmolyte. Moreover, to study the plasticity of the cells and their recovery from the osmotic stress, we washed out the sorbitol after the hypertonic stress event and collected cells in recovery after 30 min, 3 h, and 24 h, respectively (Fig. 6a). Following our high-throughput mapping of the subcellular proteome and phospho-proteome,

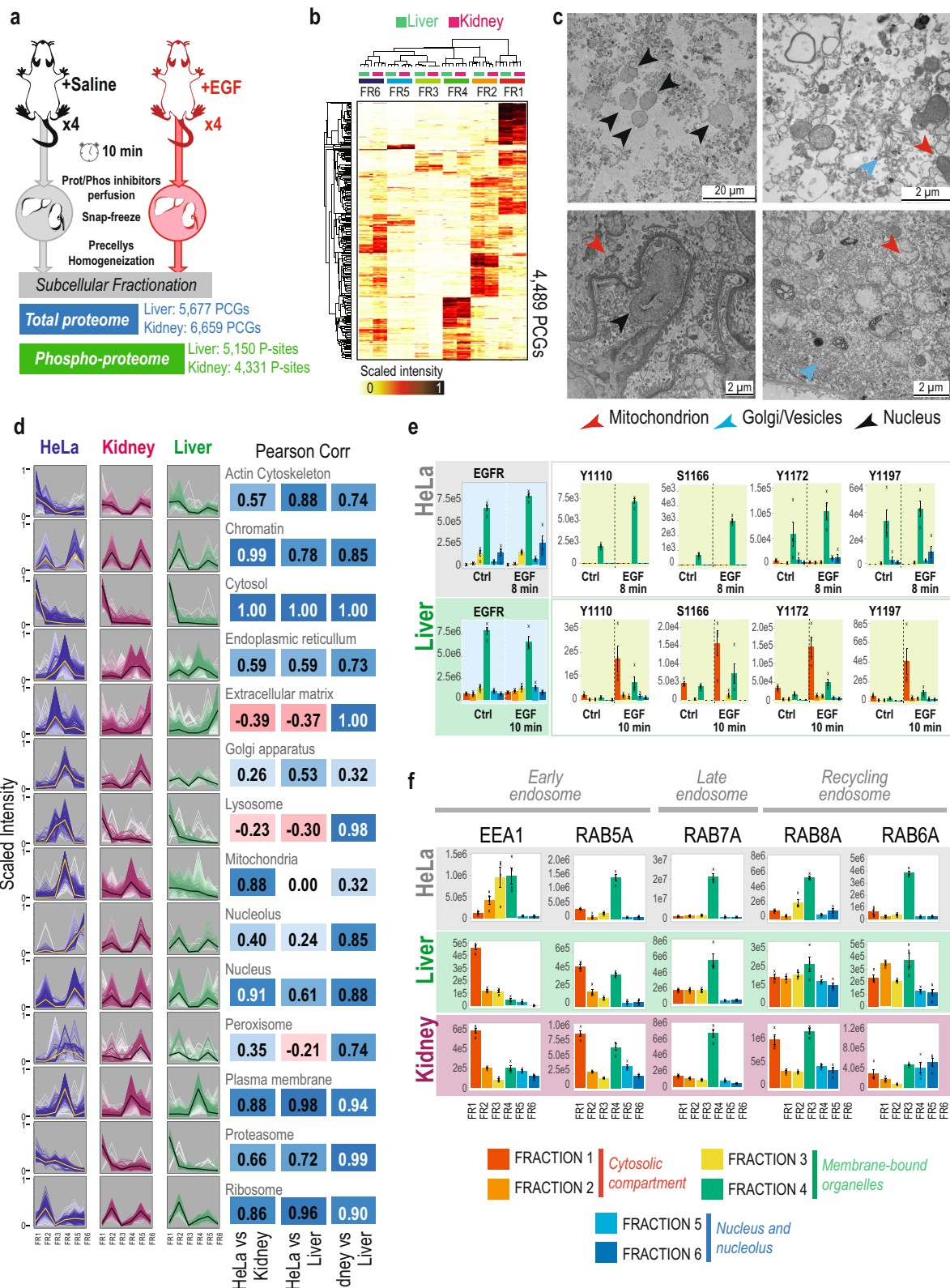

we were able to quantify 7588 proteins and 9462 phosphorylation-sites (Fig. 6a and Supplementary Data 10). To identify potential translocations as a consequence of osmotic shock, we calculated the ratios between the protein intensities after one hour of sorbitol treatment versus control conditions, for each fraction individually. We employed the ranked lists of protein ratios in each fraction at the different time points to perform Gene Set Enrichment Analysis

(GSEA)[43] using the Cellular Component Gene Ontology (GO). Interestingly, among the most significant GO terms enriched in this analysis, were the "Cytosolic Large Ribosomal Subunit (LSU)" gene set, which was significantly down-regulated in the cytosol (fraction 2) (Supplementary Fig. 9a). In contrast, we observed that the same gene set in fraction 6 (nucleus/nucleolus) inversely mirrored the trend observed in fraction 2 (Supplementary Fig. 9a),

**Fig. 5 In vivo subcellular fractionation and spatio-temporal signaling in response to EGF stimulation. a** Experimental design and workflow of subcellular fractionation proteome and phospho-proteome analysis of EGF treatment in mice and a summary of identified proteins and phosphorylation sites in liver and kidney respectively. **b** Heatmap of scaled intensities per replicate, of four replicates, of the subcellular proteome of mouse liver (green) and kidney (pink), showing both protein and sample clustering. **c** Transmission electron microscopy images of Liver (top) and Kidney (bottom) tissues after homogenization and incubation with subcellular fractionation buffer 1 containing digitonin. Blue arrows signal the Golgi apparatus, red arrows signal the mitochondrion and black arrows signal the nucleus. Sample preparation was performed in technical duplicates derived from the same organ; an aliquot at each subcellular step was pooled for TEM imaging. **d** Profile-plots of cell compartment markers in the subcellular proteome HeLa, Kidney, and Liver datasets. Scaled intensity across fractions is plotted for each independent replicate. Gradient of color indicates Pearson correlation to the centroid of each distribution, which is highlighted as a yellow/black line. Next to the plot, the Pearson correlation coefficients calculated between samples of the centroids of each cell compartment are indicated. **e** Bar-plot of protein intensities and phosphorylation sites across fractions and time points of EGFR in HeLa and Liver. Height of the bars represents the mean protein intensity of $n = 4$ experimental replicates, and error bars represent the standard error of the mean. **f** Bar-plot of protein intensities across fractions in the HeLa subcellular fractionation datasets corresponding to markers of early, late, and recycling endosomes. Height of the bars represents the mean protein intensity of $n = 4$ experimental replicates, and error bars represent the standard error of the mean. Source Data for Figs. 5e and 5f are provided as a Source Data file.

---

indicating a potential switch between the two compartments. To further evaluate this, we measured the percentage of ribosomal proteins, for small and large subunits, respectively, in each subcellular fraction at all the time points (Fig. 6b). Here, we observed that the increase of ribosomal proteins in fraction 6 was restricted to LSU proteins, in contrast to the distribution of Small Ribosomal Subunit (SSU) proteins that was unchanged. This indicate that the translocation or the accumulation in fraction 6 and depletion in fraction 2 is specific to LSU proteins only and not the whole ribosome. Importantly, the distribution reverted to original conditions, after 3 h upon stress relief, suggesting a fast recovery (Fig. 6b).

The effects of hyperosmolarity can vary depending on intensity and duration of the treatment, but it mainly involves increased cellular toxicity, for which JNK and p38 MAP kinase signaling are key effectors[44], both of which are activated by upstream MAPKKKs (i.e.: MAP3K20 or ZAK). However, not much is known about the downstream signaling elicited by hyperosmotic shock and posterior release. We observed a significant increase in p38-alpha activation loop phosphorylation (MAPK14 Y182) after 3 h of release from osmotic stress, whereas phosphorylation of JNK1 (MAPK8 Y185) peaked after 30 min of release (Supplementary Data 12 and Supplementary Fig. 9b). Interestingly, we observed rapid phosphorylation of p38 targets upon sorbitol treatment, such as STAT1-S727 (Supplementary Data 12 and Supplementary Fig. 9b), whilst JNK targets had more delayed phosphorylation kinetics with for example JUN-S63 peaking after 3 h of release (Supplementary Data. 12 and Supplementary Fig. 9b). Importantly, JNK and p38 stress signaling is known to be mediated by ZAK is response to ribotoxicity[45–47]. We found a dual distribution of ZAK (MAP3K20) between the cytosol (fractions 1 and 2) and the nucleus (fraction 5) but whilst cytosolic ZAK in fraction 2 decreased significantly after 1 h of hyperosmotic shock (Supplementary Fig. 9c, Supplementary Data 11, Fold Change Fraction 2: −0.97, q-value: 0.03), that was not accompanied by a corresponding increase in its nuclear fraction. Conversely, several ZAK phosphorylation sites, identified in the cytosolic fraction, showed a distinct up-regulation trend that peaked between 30 min and 3 h of stress relief (Fig. 6c and Supplementary Data 12). This might suggest a potential interesting connection between ZAK activation and translocation/accumulation of LSU proteins to the nucleus/nucleolus, but this would require further extensive analyses. Nevertheless, our data shows that osmotic shock induces accumulation of LSU proteins in the nucleus/nucleolus (fraction 6), and since these are the primary sites for ribosome biogenesis and assembly[48–51] we assumed that this could be a consequence of alteration in ribosome biogenesis and or assembly. To further investigate this, we extracted the subcellular proteome and phospho-proteome

information of known ribosome assembly factors[52]. As expected, all of them were purified in nuclear and nucleolar fractions (Fig. 6d). We did not observe significant changes in their protein levels or subcellular localization upon osmotic shock. However, the phosphorylation status of several ribosome assembly factors, such as NOP58 or UTP14A, changed significantly (Fig. 6d and Supplementary Data 12) indicating a functional modulation at this level due to the osmotic shock. To confirm our findings, we performed fluorescence microscopy in TIG-3 human fibroblast cell lines expressing Keima fluorescent markers for RPL10A, RPL22, RPS3, and LC3B. Fluorescent imaging revealed that upon hyperosmotic shock with 500 mM sorbitol, LSU proteins RPL10A and RPL22 showed significant accumulation in very condensed regions within the nucleus (Fig. 6e) likely representing nucleolar subcompartments. This localization was confirmed by immunofluorescence co-localization analysis of RPL10 and RPL22 with fibrillarin, a nucleolar marker (Fig. 6e).

Most importantly, such foci were not observed either for SSU protein, RPS3, or for the control (LC3B) (Supplementary Fig. 10a–c). Next, we evaluated whether the effect was purely sorbitol dependent by using decreasing concentrations of the osmolyte, and observed that the LSU protein accumulation is observed consistently with 250 mM sorbitol but not with 100 mM (Supplementary Fig. 10c). This is in agreement with a previous study showing an effect on nucleolin translocation with 200 mM sorbitol, but not with 100 mM using the same cell lines[53]. Collectively, the data suggested that hyperosomotic shock triggers certain ribotoxicity, which results in the accumulation of LSU proteins in the nucleolus, thus likely impeding proper assembly and export of the 60S ribosome to the cytosol. To pinpoint the origin of the defect leading to the observed accumulation of LSU proteins, we performed a northern blotting analysis to investigate ribosomal RNA (rRNA) processing in which two labeled probes targeting the 18S (SSU) rRNA processing intermediates (probe "a"), 28S and 5.8S (LSU) rRNA intermediates (probe "b"), respectively. (Supplementary Fig. 11a). Overall, the blots indicate a general reduction in rRNA transcription observed by reduced 47/45S precursors when cells are treated with sorbitol (Fig. 6f and Supplementary Fig. 11b). In addition, the blot with probe "b" revealed that, the band corresponding to 12S, a precursor to 5.8S that is part of the LSU, was almost completely missing, whereas the upstream 32S precursor was slightly but significantly increased (Fig. 6f, right and Supplementary Fig. 11b). Conversely, there was no apparent relative difference between the intermediates leading to 18S (SSU) rRNA analyzed by probe "a" (Fig. 6f, left, and Supplementary Fig. 11b). In addition to the difference in relative abundance of 32S and 12S, a smear below 32S was observed with probe "b", which could indicate degradation of the precursor leading to 12S and thus explain its

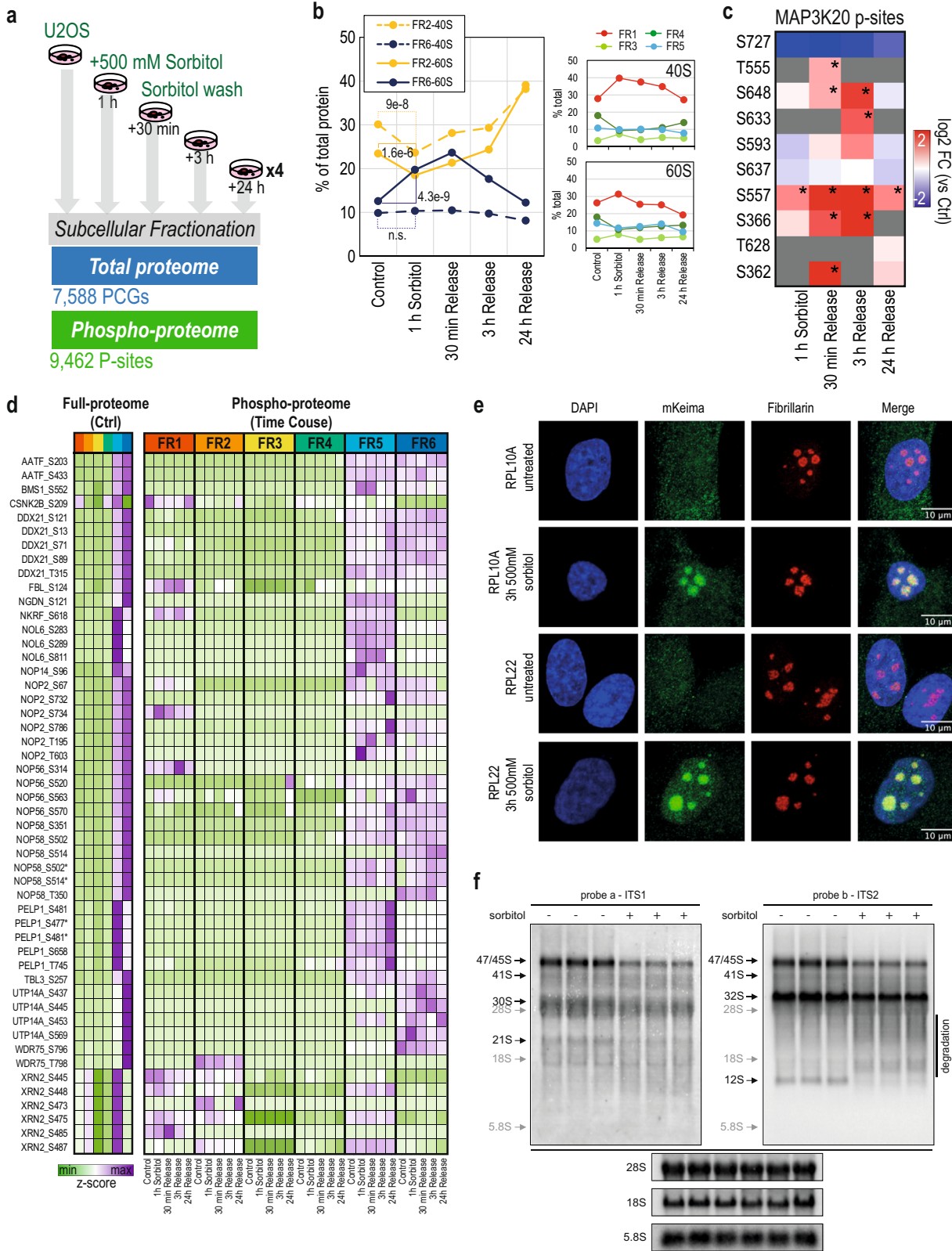

absence (Fig. 6f, right and Supplementary Fig. 11c). The analysis of rRNA biogenesis intermediates are in full agreement with nucleolar accumulation of LSU proteins specifically, most likely caused by defects in processing of 5.8S and 28S both part of the LSU. To address how sorbitol induced hyperosmotic stress mainly affects processing of LSU rRNA would require extensive analyses beyond the scope of this study.

**Muscle contraction in mice recapitulates ribosomal translocation observed in vitro upon osmotic stress.** In addition to osmotic stress, other stress stimuli elicit MAP kinase-driven activation. One such example is the mechanical perturbation during muscle-fiber contraction, which like osmotic shock, activates JNK/p38 stress signaling[54,55]. Moreover, hyperosmotic shock is, among several other effects, a known mechanical stress

**Fig. 6 Subcellular protein dynamics during hyperosmotic shock. a** Experimental design and result overview of subcellular fractionation of U2OS cells upon osmotic shock with sorbitol and posterior release. **b** Line-plot reflecting the percentage of total ribosomal protein (separately for 40S and 60S subunits) in each subcellular fraction at each given time point. Statistical significance of the change between control and 1 h treated samples is indicated by the calculated $p$ value (paired two-sample t-test). Source Data is provided as a Source Data file. **c** MAP3K20 phosphorylation sites regulation across time points. Intensity is depicted as log2 fold-change. Asterisks indicate statistical significance (moderated $t$ test, BH FDR $q$ value <0.05). **d** Heatmap of protein and phosphorylation site z-score intensities of ribosome assembly factors. Full-proteome intensity is only shown for initial/control conditions. **e** Representative images of co-localization immunofluorescence analysis of ribosomal proteins and fibrillarin in TIG3 cells expressing mKeima-tagged RPL10A, RPL22 untreated and treated with 500 mM sorbitol for 3 h. Immunostaining was performed once in two different cells (mKeima-tagged RPL10A and RPL22). **f** Northern blots of whole-cell RNA from biological replicates of U2OS cells treated with and without 500 mM sorbitol ($N = 3$), probed with probe a targeting ITS1 (left) and probe b targeting ITS2 (right). Black arrows indicate rRNA processing intermediates (see Supplementary Fig. 11a for a processing scheme) and gray arrows mark the migration of mature rRNA species. Internal RNAs (18S, 5.8S, and 28S rRNA) are employed as molecular size markers. Source Data is provided as a Source Data file.

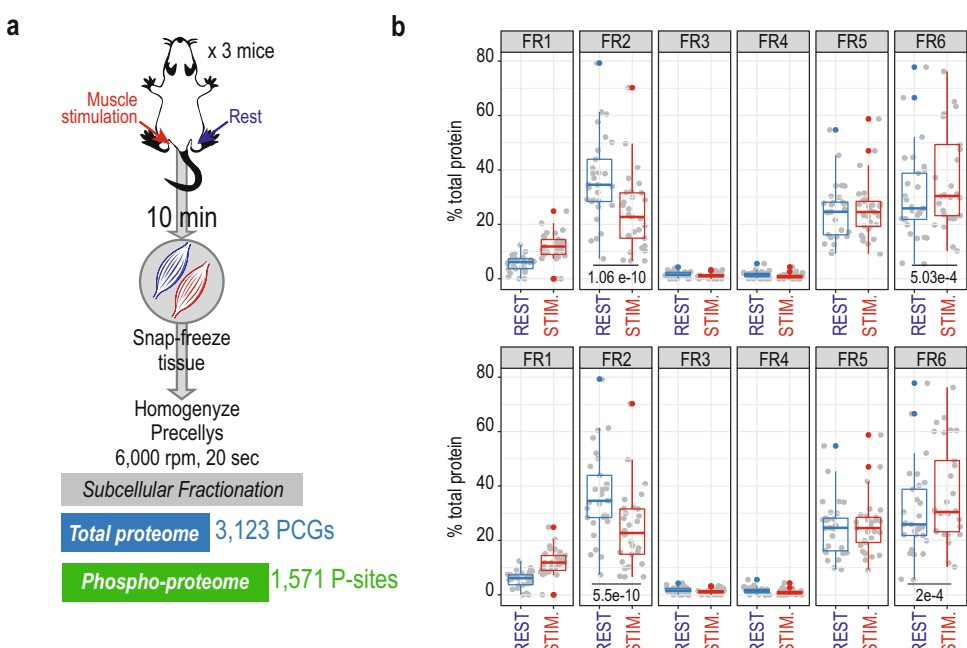

**Fig. 7 Muscle contraction in mice recapitulates ribosomal translocation. a** Experimental design and workflow of subcellular fractionation proteome and phospho-proteome analysis of muscle contraction in mice. **b** Boxplot of percentage of total ribosomal protein (top: $N = 29$ 40S subunits, bottom: $N = 37$ 60S subunits) across fractions in resting conditions (blue, $N = 3$ biological replicates) and after muscle contraction (red, $N = 3$ biological replicates). Statistical significance is calculated from a two-sided paired t-test from each subset of proteins (40S or 60S), which values were derived from 3 biological replicates in resting against stimulated mice. P-values are indicated in the figure. Boxplots show medians and limits indicate the 25th and 75th percentiles, whiskers extend 1.5 times the interquartile range from the 25th and 75th percentiles, outliers are represented by dots. Source Data is provided as a Source Data file.

insult, as it causes sudden volume compression and isotropic pressure to the cell[56]. We thus speculated that the observed ribosomal particle translocation could be caused by this mechanical perturbation. Skeletal muscle is an organ where large-scale mechanical stress can be induced experimentally through electrical nerve stimulation. To do so, we performed animal experiments by exposing one of the lower hindlimbs of anaesthetized mice to a 10 min in situ contraction protocol, while the contralateral leg served as a resting control. This was followed by the immediate harvesting of tibialis anterior (TA) muscles from both legs, which were snap-frozen and cryo-pulverized before subcellular fractionation (Fig. 7a). Due to the minute amount of sample available and the well-known high dynamic range of the skeletal muscle proteome, the coverage of the subcellular proteome and phospho-proteome was limited to 3123 proteins and 1571 phosphorylation-sites (Supplementary Data 13). However, we were able to confidently identify 37 members of the 60S ribosomal subunits and 29 members of the 40S ribosomal subunits (~80% of total ribosomal proteins), which allowed us to

evaluate location and dynamics of this organelle after mechanical contraction of the muscle. In fact, analogous to the osmotic stressed cells, we observed a similar dual distribution of the ribosomal subunit proteins between fraction 2 and fractions 5 and 6. Importantly, we confirmed that muscle stimulation also altered the distribution of the ribosomal subunit proteins, which significantly decreased in fraction 2, whilst increasing in fraction 6 (Fig. 7b), confirming the trend observed in vitro. Interestingly in muscle, we observed that the translocation was also true for the 40S ribosomal subunit proteins. Altogether, it indicates that mechanical stress recapitulates to certain extent the ribosomal translocation observed in vitro, but for the whole ribosomal complex, and not limited to 60S subunits.

## Discussion

Here, we describe a fast and high throughput approach to study spatio-temporal dynamics of the proteome and the phospho-proteome. Our workflow incorporates a simple and

straightforward sequential cell fractionation protocol to profile six subcellular compartments with high reproducibility and scalability. For high-throughput analysis, we take advantage of recent developments in MS-based proteomics methods, such as the high-field asymmetric waveform ion mobility spectrometry (FAIMS) Pro interface and the Orbitrap Exploris 480 mass spectrometer to increase proteome coverage in short liquid chromatographic gradients for single-shot runs[25]. Moreover, we also make use of a DIA approach, which circumvents certain limitations of more traditional DDA workflows for short LC-MS/MS runs, such as undersampling, dynamic range, and reduced sensitivity. In addition, we employed a spectral library-free approach (i.e. directDIA in Spectronaut), which eliminates the necessity of spending MS-acquisition time to generate spectral libraries required for DIA-based (phospho)proteomics[57]. Collectively, the optimized analytical workflow minimizes the time required to obtain comprehensive maps of the subcellular proteome and phospho-proteome dynamics, which to date was a great limitation to current MS-based spatial proteomics approaches (Supplementary Table 1). Thanks to the reduction in time to map a whole proteome and its corresponding phospho-proteome of a sample in just 5 h of MS time, we were able to apply the workflow to multiple biological replicates, cell states, stimuli, and treatment time-points. This was not only to demonstrate the high reproducibility of the method, but also to employ it to study spatio-temporal dynamics in response to cellular phospho-signaling networks activated by growth factors and stress.

We applied this chemical fractionation approach to show how EGFR adaptor proteins rearrange their subcellular distribution in response to EGF stimulation, after which they relocate from their free cytosolic form to the membrane fraction and EGFR-bound vesicles, and that this trend followed the phosphorylation activation of tyrosine residues in EGFR. However, this experiment reflected one noticeable caveat of the current workflow, which is the incapacity to separate vesicles from plasma membrane proteins in the fourth fraction when applied to epithelial cell lines. This limitation hinders us to properly track the translocation of EGFR from the cell membrane to the endosome. One possible way to solve this could be to apply surface protein labeling[58] before the subcellular protocol is performed, such that surface proteins could be separately purified from fraction 4.

Importantly, we described the applicability of this workflow to rodent tissues opening the possibility to study spatio-temporal phospho-signaling regulation in in vivo systems. However, we observed relevant differences in the cellular compartment profiles obtained from the liver or kidney when compared to HeLa cells. This suggests that differences in the nature of the sample have a significant impact in their separation properties with this workflow. Interestingly, we observed a striking difference in mitochondrial protein distribution between liver and kidney. For the kidney, we found that most mitochondrial protein markers distributed very similarly to their HeLa cell profiles, whereas it seemed that the same mitochondrial proteins were purified in earlier fractions for the liver. We used electron microscopy to explore how tissues respond to the application of this protocol, to explain the differences observed, not only between the liver and kidney, but also between tissues and cells. We found that cell integrity is disrupted but organelles remain intact, allowing the subcellular fractionation to be preserved. However, when comparing kidney to liver samples in TEM, it is clear that kidney retain certain ultrastructure, while the liver is completely disrupted by the homogenization step (Fig. 5c). Whilst, in the kidney, mitochondria, and other organelles, are contained within bigger compartments, in the liver they are released early in the process (Fig.5c and Supplementary Fig. 8b–c), exposing them to the effect of digitonin, leading to potential leakage of

mitochondrial proteins in the first purification steps and therefore a less resolute profile. To maintain cellular integrity, it would be required to perform single-cell dissociation of tissues but this is a lengthy and challenging process, which has been reported to affect gene expression[59]. Our aim in the present work is to study signaling pathways mediated by phosphorylation cascades, therefore, employing those protocols can seriously hamper the outcome of such analysis, since phosphorylation is a labile modification that can easily be degraded. To best preserve the in vivo state of the phosphoproteome, we snap froze the organs upon collection and homogenized them rapidly afterwards, but this procedure may have come at a cost of loss of cell integrity. Nevertheless, we still show the potential of this approach to study spatial regulation of phosphorylation in tissues, and we predict that optimization of single-cell suspension from tissue protocols could be adapted for an even better output of our subcellular fractionation approach.

Finally, we have demonstrated how our approach can be applied to discover previously undescribed mechanisms of the cellular stress response. Although it was already known that hypertonicity can induce ribotoxic stress due to p38 activation, we have shown by using MS-based spatial proteomics that this ribotoxicity is impacting ribosome biogenesis and assembly resulting in accumulation of 60S subunits in the nucleolus. This may be explained by the translocation of nucleolin away from the innermost fibrillar core of the nucleolus, as nucleolin is required for rRNA transcription[60], which is in line with the observed defects in rRNA processing machinery specific to 60S ribosomal subunit that we identified. Collectively, our in vivo and in vitro datasets represent a large resource of subcellular (phospho)-proteome dynamics. To make it available for other researchers in an easy accessible form, we have created a web-database SpatialProteoDynamics.github.io with a simple user interface that allows researchers to query our database of subcellular proteome and phospho-proteome dynamics for their protein of interest. Altogether, this manifests the usefulness of the methodology hereby presented for prospective studies of spatio-temporal regulation using MS-based proteomics.

## Methods

**Ethics declarations**. Mice experiments carried out in this study were performed according to the guidelines of the Danish Animal Welfare Act and the Directive 2010/63/EU for the protection of animals used for scientific purpose. The study was approved by the Institutional Animal Care and Use Committee of the University of Copenhagen and the Animal Experiments Inspectorate, Ministry of Environment and Food, Denmark (License number 2020-15-0201-00508 and 2019-15-0201-01659) and the Institutional Animal Care and Use committee of the University of Copenhagen (Project number P20–372 and P19–342).

**Statistics and reproducibility**. No sample size calculation was performed. Number of replicates was chosen based on previous expertise to obtain enough statistical power. All proteomics experiments were performed in replicates. Four biological replicates (either four cell dishes) were employed for HeLa and U2OS. For liver and kidney analysis four mice were used per condition. For muscle contraction experiments, three mice were used. In the proteomics experiments, the number of replicates was chosen to show reproducibility during the sample preparation of the subcellular fractions using the protocol described in the paper and with the aim to obtain enough statistical power. Further information about statistical analysis is provided in the methods sub-section ´Data Analyis´. No data were excluded from these analyses.

**Buffer preparation for subcellular fractionation**. The subcellular fractionation protocol requires the preparation of the following washing buffers: (i) washing solution A (30 mM Hepes pH 7.4; 15 mM NaCl, 2 mM MgCl$_2$, 1 mM EDTA), (ii) washing solution AS (30 mM Hepes pH 7.4; 15 mM NaCl, 2 mM MgCl$_2$, 1 mM EDTA, 350 mM sucrose) and (iii) washing solution AG (30 mM Hepes pH 7.4; 15 mM NaCl, 2 mM MgCl$_2$, 1 mM EDTA, 20% glycerol). Just before starting the procedure, protease and phosphatase inhibitors were added to each buffer to get the following final concentrations: 1 mM TCEP, 1 mM NaF, 1 mM beta-glycerol phosphate, and 5 mM of sodium orthovanadate. Additionally, one tablet of

cOmplete™ Mini, EDTA-free Protease Inhibitor Cocktail was added to 10 ml of the washing buffers.

**Cell culture and collection.** U2OS (ATCC HTB-96) and HeLa (ATCC CCL-2) cells were grown in a P15 dish until 70–80% confluence. Cells were serum-starved overnight. HeLa cells were stimulated for the indicated time points with 100 ng/mL of EGF. U2OS were stimulated with 500 mM Sorbitol. After 1 h of sorbitol treatment, cell medium was exchanged to wash out the sorbitol, and cells were collected at the indicated time points. Cells were washed with PBS and harvested by trypsinization (1.5 ml of trypsin). Trypsinized cells were resuspended in 8.5 ml ice-cold PBS containing 5 mM of Sodium-orthovanadate for a total volume of 10 ml and centrifuged for 3 min at 400 g. Cell pellets were washed twice with ice-cold PBS containing 5 mM of Sodium-orthovanadate. All subsequent steps were performed at 4 °C.

**Subcellular fractionation.** Cell pellets were resuspended in 540 µl of AS wash and 60 µl of 0.15% digitonin solution, which has been previously heated at 95 °C for 5 min. Samples rotated on ice for 30 min and were spun down for 3 min at 500 g in a swing-out rotor centrifuge. The supernatant was recovered and transferred to a clean tube labeled as Fraction 1. Cell pellets were washed twice with 1 ml of AS wash.

Cell pellets were resuspended in 540 µl of AS wash and 60 µl of 1.4 M NaCl. Samples rotated on ice for 15 min and were spun down for 3 min at 500 g in a swing-out rotor centrifuge. The supernatant was recovered and transferred to a clean tube labeled as Fraction 2. Cell pellets were washed twice with 1 ml of AS wash.

Cell pellets were resuspended in 570 µl of AS wash and 30 µl of 10% Tween-20. Samples rotated on ice for 15 min and were spun down for 3 min at 500 g in a swing-out rotor centrifuge. The supernatant was recovered and transferred to a clean tube labeled as Fraction 3. Cell pellets were washed twice with 1 ml of AS wash.

Cell pellets were resuspended in 540 µl of AG wash and 60 µl of 10% dodecyl maltoside. Samples rotated on ice for 15 min and were spun down for 3 min at 500 g in a swing-out rotor centrifuge. The supernatant was recovered and transferred to a clean tube labeled as Fraction 4. Cell pellets were washed twice with 500 µl of AG wash.

Cell pellets were resuspended in 540 µl of A wash, 60 µl of 5 M NaCl and 1 µl of Benzonase® Nuclease. Samples rotated on ice for 15 min and were spun down for 3 min at 2000 g in a fixed angle rotor centrifuge. Supernatant was recovered and transferred to a clean tube labeled as Fraction 5. Cell pellets were washed once with 500 µl of AG wash.

Cell pellets were resuspended in 522 µl of A wash, 60 µl of 1.4 M NaCl and 18 µl of 10% SDS. Samples were boiled for 10 min at 95 °C. Vials containing this last fraction were labeled as Fraction 6.

All collected fractions were spun in a fixed angle rotor centrifuge for 5 min at 20,000 g and transferred to clean Eppendorf tubes. Samples were stored at −80 °C for further analysis.

**Mice EGF stimulation and tissue collection.** Littermate male C57BL/6JRj mice obtained from Janvier Labs (Le Genest-Saint-Isle, France) at six weeks of age were housed at the animal facility of the University of Copenhagen in individually ventilated static type II cages (Techniplast) with access to food (Altromin 1314, Altromin) and water ad libitum and a controlled temperature and relative humidity environment ($22 \pm 2$°C and $55\% \pm 10\%$, respectively) with 12:12 h dark:light cycle.

For EGF stimulation experiments, adult mice (eight weeks age, $22.1 \pm 2.3$ g weight) were assigned to two study groups of four mice each by simple randomization. Mice were anesthetized with 2% isoflurane in oxygen using a precision vaporizer (Leica Biosystems). In each group, four mice were administered sterile epidermal growth factor in isotonic saline (EGF, 100 µg/kg bodyweight) or isotonic saline intravenously in a single bolus dose into the inferior vena cava. 10 min post injection, the animals were perfused (1.5 min, 4.5 ml/min) with ice-cold isotonic saline containing protease inhibitors (Roche cOmplete™ Mini, EDTA-free Protease Inhibitor Cocktail) and phosphatase inhibitors (1 mM NaF, 1 mM beta-glycerol phosphate and 5 mM of sodium orthovanadate) using a syringe pump (Aladdin AL-1000, World precision instruments). Livers and right kidneys were quickly removed and snap frozen in liquid nitrogen. The total time from dosing to tissue collection was 12 min. For subcellular fractionation, only part of the median lobe of the liver was used.

**Mice muscle stimulation and tissue collection.** For muscle stimulation, fed mice were anesthetized by an intraperitoneal injection of pentobarbital (10 mg/100 g body weight, diluted 1:10 in a 0.9% saline solution) and left to recover on a heating plate (30 °C) for ~20 min. Subsequently, an electrode was placed on a single common peroneal nerve followed by 10 min in situ contraction of TA muscle. The contralateral leg served as a sham-operated resting control. The contraction protocol consisted of 0.5 s trains repeated every 1.5 s (frequency: 100 Hz; duration; 0.1 ms; voltage; 5 V). TA muscle from both legs was removed immediately following euthanasia and snap frozen in liquid nitrogen.

Prior to subcellular fractionation, tissue samples were homogenized using a Precellys system with 3 beads (2.8 mm) for 20 s at 6,000 rpm in 1 ml buffer A. After centrifugation at 2000 g, the supernatant was removed. Sample was washed twice in 500 µl buffer A before starting the subcellular fractionation.

**SDS-PAGE sample preparation.** The subcellular fractionation protocol was carried out on HeLa cells. The six fractions and the washes were collected. Each fraction and washes were concentrated using Sartorius Filtrate Tube 10 KDa. Buffer was exchanged to 50 mM Tris, and samples were concentrated up to 50 ul. 50 % in volume of the fractions and the washes were denatured in NuPAGE LDS Sample Buffer (4X) and reduced in DTT for loading on polyacrylamide gels (1.0 mm×10 well NuPAGE 4–12% Bis-Tris Gel). The gel was stained overnight with Instant-BlueTM (Sigma-Aldrich) and rinsed twice with water. The most intense bands per fractions were cut out, shred into smaller pieces and destained by repeating 4 times successive washes with 50 mM ABC and 50 mM ABC/ACN (50/50 v/v) before being dried with 100%ACN, each time with 10 min incubation at 25 °C. The proteins were reduced with 15 mM TCEP and alkylated with 30 mM CAA. Samples were digested overnight in-situ using trypsin. The digestion was quenched by adding 1% TFA, final concentration. One fourth of the extracted peptide samples were loaded on an Evotip and injected for LC-MS/MS analysis.

**Sample preparation for MS analysis.** Subcellular fractions were denatured, reduced, and alkylated with 0.3% SDS, 5 mM TCEP and 10 mM CAA during 10 min at 95 °C. Afterwards, samples were digested overnight using the PAC protocol[24] implemented for the KingFisher™ Flex robot (Thermo Fisher Scientific) in 96-well format as described previously[25,61]. Samples were divided into two wells for digestion, such that 300 µl of sample were digested in parallel. The 96-well comb is stored in plate #1, the sample in plate #2 in a final concentration of 70% acetonitrile and with 50 µl of magnetic Amine beads (ReSyn Biosciences) in a protein/bead ratio of 1:2. Washing solutions are in plates #3–5 (95% Acetonitrile (ACN)) and plates #6–7 (70% Ethanol). Plate #8 contains 300 µl digestion solution of 50 mM ammonium bicarbonate (ABC), 0.5 µg of LysC (Wako) and 1 µg trypsin (Sigma Aldrich). Protein aggregation was carried out in two steps of 1 min mixing at medium mixing speed, followed by a 10 min pause each. The sequential washes were performed in 2.5 min and slow speed, without releasing the beads from the magnet. The digestion was set to 12 h at 37 degrees with slow speed. Samples were acidified after digestion to final concentration of 1% trifluoroacetic acid (TFA). 20 µl of each sample were loaded directly into Evotips (Evosep) for full proteome analysis. Remaining sample was loaded onto Sep-Pak cartridges (C18 1 cc Vac Cartridge, 50 mg - Waters).

**Phosphopeptide-enrichment of subcellular fractions.** In order to perform phosphopeptide-enrichment of each subcellular fraction, peptides previously loaded into Sep-Pak cartridges were eluted into the KingFisher™ Flex robot (Thermo Fisher Scientific) plate using 75 µl of 80% ACN. 150 µl of loading buffer (80% ACN, 8% TFA and 1.6 M glycolic acid) was added to each sample. Phosphopeptide-enrichment was performed as described previously[25,61] using 10 µl of TiIMAC-HP beads (MagResyn, Resyn Biosciences). Briefly, the 96-well comb is stored in plate #1, 10 µl Ti-IMAC HP beads in 100% ACN in plate #2 and loading buffer (1 M glycolic acid, 80% ACN, 5% TFA) in plate #3. Sample is eluted from Sep-Pak cartridges with 75 µl of 80% ACN and completed with 150 µl of 1.6 M glycolic acid, 80% ACN, 8% TFA and added in plate #4. Plates 5–7 are filled with 500 µl washing solutions; loading buffer, 80% ACN, 5% TFA, and 10% ACN, 0.2% TFA respectively. Plate #8 contains 200 µl 1% ammonia for elution. The beads were washed in loading buffer for 5 min at medium mixing speed, followed by binding of the phosphopeptides for 20 min and medium speed. The sequential washes were performed in 2 min and fast speed. Phosphopeptides were eluted in 10 min at medium mixing speed. Eluted phosphopeptides were acidified with 10% TFA to pH <3 and loaded into Evotips (Evosep) for further MS analysis.

**LC-MS/MS analysis.** All samples were analyzed on the Evosep One system using an in-house packed 15 cm, 150 µm i.d. capillary column with 1.9 µm Reprosil-Pur C18 beads (Dr. Maisch, Ammerbuch, Germany) using the pre-programmed gradient for 60 samples per day. The column temperature was maintained at 60 °C using an integrated column oven (PRSO-V1, Sonation, Biberach, Germany) and interfaced online with the Orbitrap Exploris 480 MS (Thermo Fisher Scientific, Bremen, Germany) using Xcalibur (tune version 1.1). When using FAIMS, spray voltage was set to 2.3 kV, otherwise it was set to 2 kV, funnel RF level at 40, and heated capillary temperature at 275 °C. For full-proteome analysis of subcellular fractions using DIA and FAIMS full MS resolutions were set to 120,000 at m/z 200 and full MS AGC target was 300% with an IT of 45 ms. Mass range was set to 350 − 1400. AGC target value for fragment spectra was set at 100%. 49 windows of 13.7 m/z scanning from 361 to 1033 m/z were used with an overlap of 1 Da. Resolution was set to 15,000 and IT to 22 ms and normalized collision energy was 27%. Compensation voltage for FAIMS was set to −45. For phospho-proteome analysis using DIA we employed 17 windows of 39.5 m/z scanning from 472 to 1143 m/z with 1 m/z overlap. Resolution was set to 45,000 and IT to 86 ms. Normalized collision energy was set at 27%. All data were acquired in profile mode using positive polarity.

**Raw data processing**. Full proteome and phospho-proteome subcellular fraction raw files were searched using Spectronaut (v14) with a library-free approach (directDIA) using either human database (Uniprot reference proteome 2019 release, 21074 entries) or mouse database (Uniprot reference proteome 2019 release, 22286 entries), supplemented with a database of common contaminants. Carbamylation of cysteines was set as a fixed modification, whereas oxidation of methionines and acetylation of protein N-termini were set as possible variable modifications. Additionally, for phospho-proteome analysis, and phosphorylation of serine, threonine and tyrosine were included as well. The maximum number of variable modifications per peptide was limited to 3. Only for phospho-proteome files, PTM localization cutoff was set as 0.75. Cross-run normalization was turned off. For protein quantification, major protein group aggregation method was changed to sum. Phospho-peptide quantification data was exported and collapsed to site information using the Perseus plugin described in Bekker-Jensen et al (see Code Availabitiy)[57]. All remaining processing steps were performed in either Perseus (v1.6.5.0) or R (v3.6.2).

Raw MS files from the in-gel digestions were processed in Spectronaut (v15) with same settings as before. Peptide quantification of gel bands was performed comparing MS1 TIC alignment across the samples.

**Data Analysis**. Data at protein and phospho-site level were processed using R (v3.6.2). For normalization, to remove experimental bias, as well as for imputation of missing values, each fraction was treated separately. Protein identifications without valid gene names were discarded. Data was log2 transformed and three valid values in at least one experimental group were required to preserve the protein or phospho-site. Most of the data analysis was performed using functions implemented in the Dapar package (v 1.18.3)[62] and following the data analysis pipeline of Prostar (v 1.18.4)[62]. Normalization was performed using loess function from limma package (v3.42)[63]. Imputation of missing values was performed in two steps taking into account the nature of the missing values, as described by Lazar et al[64]. First, we considered partially observed values or 'Missing Completely at Random' as those values missing within a condition in which there are valid quantitative values in other replicates. These partially observed values were imputed using the KNN function (at protein level) and slsa function (at phospho-site level). Secondly, values missing in an entire condition, also termed as 'Missing Not at Random' were imputed using the detQuant function from imp4p package (v0.8), which impute the values using a constant low value calculated from a given quantile threshold (quantile = 2.5). Whilst the first approach is based in similar protein/peptide expression profile to estimate the value to use to impute the missing intensities, the second type of missing values imputation assumes that the protein/peptide could not be quantified because its intensity was below detection levels or just not present in the sample.

Finally, differential expressed protein and sites were calculated using limma (two-sided, BH FDR < 5%), requiring at least three valid values in one of the two experimental conditions compared.

For all barplots, error bars indicate standard deviation of the mean of four replicates.

**Identification of translocation events in EGF stimulation time course**. First, "Mobility Score" was calculated to rank the proteins according to how much their profiles change between different time points (Supplementary Fig. 12a). To do so, we scaled the protein levels for each fraction at a given time-point to the total abundance, which results as summing all fractions. Then we calculated the absolute difference between fractions at each time point against the control or initial condition, which represents the percentage of the protein that changes distribution. Then, the two compartments that show the highest difference are selected, since those would be the ones between which the protein likely moves. Based on that potential translocation events in four categories: cytosol-nuclear (blue), cytosol-membrane (green), membrane-nuclear (red) and within the same neighborhood (grey), to make the visual inspection of the plots easier.

Secondly, in order to confidently identify a "moving" protein, it is required to change in abundance in both compartments (i.e., if a protein moves from the cytosol to the membrane, its intensity should decrease in the cytosolic compartment and increase in the membrane one). For that, we calculated the significance of that change as the combination of the p-value from the moderated t-test performed in each compartment at each time point (i.e., p-value from FR1 2 min vs FR1 Control and p-value from FR4 2 min vs FR4 Control).using the Fisher's method, followed by correction for multiple testing by Benjamini-Hochberg (Supplementary Fig. 12b).

Finally, the movement score and the combined p-value are plotted and proteins showing a movement score >0.1 (or 10%) and a combined p-value (FDR corrected) <0.05 are classified as potential targets for translocation (Supplementary Fig. 12c).

**MetaMass analysis**. The precision of partitioning of proteins in subcellular compartments was assessed using an updated version of the Excel-based application MetaMass (Supplementary Data 4)[22]. A detailed user manual for the published version can be retrieved from the Nature Methods website (https://www.nature.com/articles/nmeth.3967#Sec12). Briefly, the input is a list of proteins assigned to groups by k-Means clustering of a dataset, or a combination of multiple datasets.

The output includes assigned locations for each protein with scores for reliability (precision) and statistics for recall and precision for the proteins used as markers for subcellular compartments.

The user pastes the list of protein and assigned groups into the spreadsheet and clicks "buttons" to select among several built-in sets of markers for subcellular locations. Some sets correspond to single-location annotations from Uniprot, The Gene Ontology Consortium and the Compartments Database (all retrieved January 2021). Others are locations assigned in spatial proteomics studies[9,30]. MetaMass assigns proteins within a given group to the same subcellular location based on the content of marker proteins. For example, if two proteins in the group are markers for cytosol, and none are markers for other locations, all proteins in that group are assigned to the cytosol with a precision of 1. If the group also contains a marker for e.g. nucleus, the proteins are still assigned to cytosol, but with a precision of 0.66. If all markers for a given location are assigned to the correct compartment, the precision and recall for that compartment is 1. E.g., if a third is assigned to the wrong locations, the precision is 0.66.

MetaMass II has a wider range of marker sets than what was included in the published version. There are also data from recent spatial proteomics studies to facilitate meta-analysis. All datasets were normalized to the maximum signal value measured across the fractions. The classification is based on standard Excel functions, and the functions are displayed by selecting the cells. The worksheets are protected to prevent the user from accidentally editing cells that contain formulas. The password to unprotect the sheets to make modifications is "1". The workbook has macros to automate the analysis. Experienced Excel users will know how to view and modify the codes. We have included a more thorough guide for using MetaMass in the Supplementary Note 2.

**Northern blotting analysis of ribosomal RNA processing**. Ten μg whole cell RNA from U2OS cells treated with and without 500 mM sorbitol for 3 h was separated on a formaldehyde denaturing 1% agarose gel. Then transferred to a BrightStar-plus nylon membrane (Ambion) by capillary blotting, followed by UV-cross-linking. Probes (10 pmol each) were labeled with [γ-32-P]-ATP using T4 PNK (Thermo Fisher) and hybridized to the membrane one by one in hybridization buffer (4× Denhardts solution, 6× SSC, 0.1% SDS) overnight at 10 °C below the Tm of the probe. The membrane was washed four times in washing buffer (3× SSC, 0.1% SDS), exposed to a Phosphor Imager screen, scanned by a Typhoon scanner (GE Healthcare) and analyzed using Fiji software (v1.53c). The membrane was stripped between each hybridization using boiling hot 0.1% SDS.
Probe a; TGGGTGTGCGGAGGGAAGC
Probe b; ACGCCGCCGGGTCTGCGCTTA
28 S rRNA; GCTCCCGTCCACTCTCGAC
18 S rRNA; CCAGACAAATCGCTCCACCAACTAAG
5.8 S rRNA; CCGCAAGTGCGTTCGAAGTGT

**Generation of stable cell lines**. mKeima fusion sequences were cloned into pLVX-TetOne-Puro backbone using In-Fusion or Gibson Assembly cloning kits according to manufacturer's instructions. RPL10A, RPS3 and LC3B have an N-terminal mKeima tag, while RPL22 carries a C-terminal mKeima tag. Stable cell lines were generated by lentiviral transduction of TIG3 fibroblasts. To generate virus, HEK293 (ATCC CRl-1573) cells were transfected with these plasmids along with PAX8 and VSV-G expressing plasmids and the virus collected after 24 h. After transduction, positive cells were selected with puromycin and subsequently FACS sorted for keima expression.

**Cell culture, treatments, and Keima imaging**. TIG-3 fibroblasts were a gift from Anders Lund's lab at Biotech Research & Innovation Center, University of Copenhagen. TIG3 fibroblast were grown in Dulbecco's Modified Eagle Medium supplemented with 10% Fetal Bovine Serum and 1% PenStrep. Keima expression was induced for 72 h before sorbitol treatment with 100 ng/ml doxycyclin. Sorbitol was dissolved in media immediately before being added to the cells. Cells were imaged live in HBSS with 1:6000 Hoechst (H3570) using an ImageXpress Micro Confocal High-Content Imaging System (excitation 440 nm, emission 620 nm). Analysis of nuclear keima puncta was performed with the MetaXpress software. Nuclei were segmented based on Hoechst stain and keima positive puncta with a minimum width of 1μm and maximum width of 8μm, and with pixel intensity of minimum 3000 above local background were quantified within the nucleus.

**Co-localization of keima and fibrillarin**. TIG3 cells were treated with 500 mM sorbitol for 3 h and fixed in 4% formaldehyde for 10 min. Fixed cells were incubated for 1 h at RT in IF buffer (3% BSA, 0.1% Triton-X) with primary antibodies: keima anti-mouse (MBL M126−3M, 1:200) and fibrillarin anti-rabbit (abcam ab5821, 1:200). Cells were washed 3x in PBS and incubated for 1 h at RT in IF buffer with secondary antibodies: Alexa Fluor rabbit-568 (Thermo Fisher A10042, 1:1000) and Alexa Fluor mouse-488 (Thermo Fisher A11029, 1:1000). Cells were stained with DAPI (Sigma-Aldrich 10236276001) for 10 min and mounted in DAKO mounting medium (S3023). Cells were imaged with a Zeiss LSM800 confocal microscope with 63x/1.4 oil DIC objective using Zeiss Zen software.

**Electron microscopy**. Pellets with Hela cells, liver and kidney homogenate constituting the starting materials and pellets formed by centrifugation after each step in the fractionation protocol were fixed with 2% v/v glutaraldehyde in 0.05 M sodium phosphate buffer (pH 7.2). The pellets were then embedded in agarose, rinsed three times in 0.15 M sodium phosphate buffer (pH 7.2), and subsequently post-fixed in 1% w/v $OsO_4$ with 0.05 M $K_3Fe(CN)_6$ in 0.12 M cacodylate buffer (pH 7.2) for 2 hrs. The specimens were then dehydrated in graded series of ethanol, transferred to propylene oxide and embedded in epon (812 Resin kit, TAAB Laboratories Equipment Ltd, Aldermaston, UK) according to standard procedures.

Sections, approximately 60 nm thick, were cut with an Ultracut 7 (Leica, Vienna, Austria) and collected on formvar coated copper grids (Electron Microscopy Sciences, Hatfield, PA), stained with 0.5 % w/v uranyl acetate and 3 % w/v lead citrate. Subsequently, they were examined with a Philips CM 100 Transmission EM (Philips, Eindhoven, The Netherlands), operated at an accelerating voltage of 80 kV. Digital images were recorded with an OSIS Veleta digital slow scan 2k x 2k CCD camera (Olympus, Münster, Germany) and the iTEM software package.

**Reporting summary**. Further information on research design is available in the Nature Research Reporting Summary linked to this article.

## Data availability
The mass spectrometry proteomics data have been deposited to the ProteomeXchange Consortium via the PRIDE partner repository with the dataset identifier PXD023690. Source data are provided with this paper. Spatio-temporal proteomics data acquired in this project can be explored through https://SpatialProteoDynamics.github.io. Source data are provided with this paper.

## Code availability
Custom R code used in the manuscript is available in the GitHub at https://github.com/SpatialProteoDynamics/SpatialProteoDynamics.github.io and via Zenodo[65] at https://doi.org/10.5281/zenodo.5635633. PTM collapse plugin requires Perseus and R (minimum version 3.6.0) to run and it is available at https://github.com/AlexHgO/Perseus_Plugin_Peptide_Collapse.

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

## Acknowledgements

The authors would like to thank Jørgen F.P. Wojtaszewski (University of Copenhagen, Denmark) for access to in situ muscle contraction equipment. Miguel Garrido Rodrigo for his help setting up the web-based resource. We acknowledge the Core Facility for Integrated Microscopy (CFIM), Faculty of Health and Medical Sciences, University of Copenhagen, for help with sample analysis by transmission electron microscopy. Work at The Novo Nordisk Foundation Center for Protein Research (CPR) is funded in part by a generous donation from the Novo Nordisk Foundation (Grant number NNF14CC0001). The proteomics technology developments applied was part of a project that has received funding from the European Union's Horizon 2020 research and innovation programme under grant agreements: MSmed-686547, EPIC-XS-823839, PUSHH–861389 and ERC synergy grant 810057-HighResCells. The work was also supported by a research grant from The Novo Nordisk Foundation to A.L. (NNF20OC0059767). F.L.-J. is partner in the National Network for Advance Proteomics Infrastructure (NAPI), which has received funding from the Norwegian Research Council. T. T. is supported by a grant from the Norwegian Cancer Society to F.L.-J. Work in the Bekker-Jensen (S.B.-J) lab was supported by the European Research Council (ERC) under the European Union's Horizon 2020 research and innovation program (grant agreement 863911 - PHYRIST).

## Author contributions

A.M.-V. designed, optimized and performed all proteomics experiments, analyzed the data and wrote the manuscript. D.B.B.-J. and S.S. helped perform the cell line experiments. O.Ø. helped perform the electron microscopy analysis. C.K. performed MS and data analysis on SDS-PAGE gels. K.S., A.M., T.T. and F.L.J. developed the cell fractionation protocol. F.L.J. developed the updated version of MetaMass and performed MetaMass-based analysis. E.T.V. and A.L. performed the in vivo EGF stimulation experiments. E.K., S.B.-J., S.H.B. and L.B.F. performed fluorescent imaging experiments. R.K. performed in situ muscle contraction. N.K performed the rRNA processing experiment. J.V.O. designed the experiments, critically evaluated the results, analyzed the data, and wrote the manuscript. All authors read, edited, and approved the final version of the manuscript.

## Competing interests

D.B.B.-J. is employee at Evosep Biosystems. All other authors declare no competing interests.
