## [Peer Review File · Nature Communications]

REVIEWER COMMENTS

Reviewer #1 (Remarks to the Author):

In the manuscript entitled “Spatial-proteomics reveal in-vivo phosphor-signaling dynamics at subcellular resolution” Martinez et al claim to develop a high-throughput workflow to rapidly fractionate and profile the proteome/phosphor-proteome at subcellular resolution. Using this workflow, they to investigate spatio-temporal proteomic/phospho-proteomic response during EGFR-phospho-signaling and osmotic stress at the cellular level, as well as during muscle contraction with the mouse model.

By sequentially treating cellular lysates to different detergents and different salt concentrations, the authors claim to chemically fractionate the subcellular contents. The contents of the six fractions were subjected to high-resolution mass spectrometry. The authors compare the performance of their workflow with a commercial kit used in a previous study (Mendes et al. 2017). As judged by mass spectrometry analyses, the protocol used in the current study seems to perform as well as the workflow reported by Mendes et al. 2017. This part of the the manuscript has been carefully performed.

In my opinion, the biological (and mechanistic) insight(s) gained through the “less cumbersome” fractionation coupled mass spectrometry workflow to investigate EGFR-phospho-signaling, and in particular the causal link between osmotic stress/muscle stimulation and ribosomal protein accumulation seems limited.

Using cell biological studies, the authors claim an accumulation of the large ribosomal subunit proteins Rpl10a and Rpl22, but not small subunit ribosomal proteins, in the nucleus/nucleolus upon osmotic stress. The data presented are of poor quality and are not of the standard of the ribosome cell biology field. It would be important for the authors to show clear co-localization with additional nucleolar/nuclear markers. The nucleolar/nuclear accumulation of ribosomal proteins Rpl10a and Rpl22 was used as an argument for stalling of 60S pre-ribosomal particles in the nucleolus. If this were true, pre-rRNA levels should have increased in response to osmotic stress. Instead, Northern analyses indicate reduced pre-rRNA levels upon osmotic stress. If pre-rRNA processing was affected at least one of the pre-rRNA species should have accumulated. Rpl10a is incorporated into the pre-ribosome only in the cytoplasm, therefore its nucleolar/nuclear location is puzzling. The position of Rpl10a on pre-ribosomes is typically occupied by assembly factors/export adaptors, therefore it seems likely that free Rpl10a mislocalizes and accumulates in the nucleus/nucleolus. The authors need to address the issue whether Rpl22 and Rpl10a are bound to pre-ribosomes, to argue the point that 60S pre-ribosomal particles are accumulating in the nucleolus. The functional importance of

accumulation of either of the ribosomal protein and/or pre-ribosome in the nucleolus upon osmotic stress and muscle contraction is descriptive. Overall, the study does not provide a fresh perspective to the links between critical anabolic processes and osmotic/muscle stress.

Reviewer #2 (Remarks to the Author):

Reviewer's Report

Title: Spatial-proteomics reveal in-vivo phospho signaling dynamics at subcellular resolution

by Ana Martinez-Val et al.

Martinez-Val and coworkers developed a “high-throughput” mass spectrometry(MS)-based method for analyzing changes in the subcellular localization of proteins and phosphoproteins to study dynamic signaling events at a subcellular resolution. The described workflow for sample generation and LC-MS/MS analysis is used for temporal studies and the tracking of dynamic changes in the (phospho)proteome at a subcellular level. Analyses of EGFR signaling dynamics in HeLa cells as well as hypertonic stress-induced protein translocation in U2OS cells serve as proof-of-concept. Protein translocation by mechanical contraction is also tracked in mouse samples and some data are presented indicating relocation of ribosomal proteins.

What is notable about this work is that it aims at determining subcellular changes in the proteome and phosphoproteome in response to a stimulus or stress condition. It therefore advances spatial proteomics to the level of the phosphoproteome. For this, the authors apply a streamlined sample preparation workflow and state-of-the art MS methodology to reduce MS-acquisition time but still cover a large fraction of the proteome and phosphoproteome. However, the chemical fractionation method they employ does limit their approach because the resolution it provides for discriminating between discrete changes in the subcellular localization of proteins and phosphoproteins is rather low. In fact, the authors also state that “this analysis revealed three major cellular compartments” when they discuss Fig. 1F. Based on the presented data (1F, Fig 2, Fig 3B etc.), this fractionation method is applicable to follow changes in protein localization between cytosolic compartments, membrane-bound organelles and nucleus/nucleolus but not between the different organelles, e.g. the ER and mitochondria (at least not at a moderate to high confidence level). The authors should present and discuss this limitation more clearly and consistently in their manuscript and, importantly, this limitation remains true regardless of the improvement of the chemical fractionation method compared to a commercial kit (Mendes et al.; Fig. 2) or use of high-end MS methodology.

One can also have different views on “high throughput” versus “resolution”. In the reviewer’s point of view this study would benefit from high(er) resolution at a subcellular level and a detailed analysis rather than performing a high number of different “proof-of-concept” studies with limited resolution

and each with less informative value. Likely due to the rather high number of different analyses the authors present in this work, the overall presentation, analysis and interpretation of the acquired datasets is partly insufficient in detail and limited in depth and thoroughness. In the literature, there is also increasing evidence that 30-50% of all proteins have dual or multiple subcellular localizations, a circumstance that is not mentioned and addressed in this work. The authors therefore might consider and assess how this is reflected in their spatial proteomics data and how does it affect data interpretation. Furthermore, to minimize the risk of overinterpretation, it is also suggested that the authors consistently apply statistical testing to each dataset and report the exact p-values to reveal whether there is a significant change in the localization of a (phospho)protein based on the acquired data or whether it is just a tendency. Independent validation of data is suggested to support claims. More specific suggestions that hopefully help the authors to improve their work are listed in the following:

Comments

1. Lines 48-51: Sentences with references 6-8. It is suggested to add further references to better represent published work.
2. Line 59: “liquid chromatography coupled to mass spectrometry (LC-MS/MS)” should be “liquid chromatography coupled to tandem mass spectrometry (LC-MS/MS)”
3. The authors just refer to “Methods” when introducing the chemical fractionation method in the result part (Line 93). However, in the Discussion they describe and discuss this fractionation method and the different reagents used in detail. For better clarity, it is suggested to already introduce in the Results the properties of the fractionation chemicals and provide more details why these were used, what they do and what has been changed in the protocol compared to previous work.
4. The authors should provide additional information on the MS-acquisition method as the statement “more sensitive MS-acquisition method” (line 113) is not very informative. In addition, abbreviations used for LC-MSMS methods (SPD, CV) in Fig. 1A should be described in the Figure legend. Since the cell lines HeLa and U2OS are shown in the workflow, this should be described in the legend, clarifying for which cell line data are shown in Fig 1B-F. “LC-MSMS” should also be “LC-MS/MS”.
5. Lines 117-121: This interpretation is not conclusive because in Fig. 1F it can be seen that nearly all marker protein profiles have low intensities in fraction 3. So, there is no subcellular compartment enriched in fraction 3.
6. It is not clear how the data was normalized. Was loess normalization applied to the complete dataframe? If so, wouldn't the assumption that the majority of proteins between samples is unaltered be violated? It would be nice if the material and methods would suffice to recapitulate the data processing. Also, it is not clear how the intensities for the profile plots in Fig. 1F were scaled.
7. The authors should comment why lysosomes and ER marker proteins have two distinct peaks in the average profile plots (Fig. 1F) and how many of these follow one of the two different profiles –

what is the difference between the two populations? Have the different marker protein lists been checked manually for correctness and proteins with multiple localizations?

8. The authors should comment on the relationship between Fig. 1D and Fig. S1B. If both show relationships between the fractions, how can it be that in the agglomerative clustering fraction 3 is closer to 5 and 6, while in Fig. S1B fraction 3 seems to have an overall higher correlation with fraction 4 and 1?

9. Lines 122-127: The experiment described has been conducted in HeLa cells. U2OS cells are first introduced in line 166. Therefore, it is not apparent why the correlation of the summed intensities of the fractions and the intensities of the whole cell lysate of the U2OS cell line is shown in Fig. S1A. It is suggested to introduce both HeLa and U2OS when referring to Fig. 1. In addition, the intensities in both approaches should increase linearly and thus the Pearson correlation coefficient would be more appropriate to use.

10. Lines 142-148: The authors refer to GO enrichment analysis and that it “revealed even more specific pattern for each fraction...” Unfortunately, these data are only presented in a Suppl. Table. GO term enrichment analysis would be indeed helpful to better judge the resolution of the fractionation approach and thus it is recommended to include the data as a plot for each fraction as in Fig. 1F.

11. Line 148: Using the Cell Atlas data as reference cannot be considered as validation of the acquired data. It is also suggested to mention in each plot in Suppl Fig. 2A how many protein profiles are shown. This also applies to e.g. Suppl Fig. 2B.

12. Lines 152-157: Please show that this enrichment in fraction 1 is significant. The authors should also clarify to which ribosomal proteins they specifically refer. The conclusion is too speculative.

13. Lines 160-162: Statement is not justified; see comment 5.

14. Lines 162-165. The authors should comment why these kinases and substrates were selected including the source of annotation and why not a more unbiased data analysis has been performed? They should also mention the subcellular localization of each of these kinases.

15. Suppl Fig. 2E-G: It is suggested to present this comparison earlier in the Results part.

16. The data obtained in this work are first compared to a published dataset using a commercial kit for subcellular fractionation, and compare favorably as judged by F-score statistics as shown in Fig. 2C. However, it is not easy to judge whether the sequential cell fractionation protocol provides a better partitioning compared to the previous approach as the analysis pipelines used greatly differ (label-based quantification, FAIMS, DDA, CID vs. label-free, DIA, HCD). Comparison of the described fractionation protocol against established protocols using the same analysis pipeline would be of interest. Furthermore, visualization of the data after dimension reduction (e.g. principal component analysis, tSNE etc.) is suggested to make the comparison more meaningful. The authors should also state that the LOPIT-based and SubCellBarCode methods clearly show better partitioning of subcellular localizations as visible from F-scores shown in Suppl. Fig. 3E.

17. Lines 184-186: It is not clear what exactly was unexpected. Were the F-scores for the membrane bound compartments unexpectedly high given that these organelles were not well separated, or does this refer to the comparison to previous results?

18. Lines 190-192: Why is the precision of centrifugation-based methods higher as expected? It is also not clear what the authors mean with "approximate locations"? The authors should be more precise here. Also from these data it is evident that the chemical fractionation approach is very good in localizing proteins to cytosolic compartments, membrane-bound organelles in general and nucleus/nucleolus but that it is considerably less effective in revealing protein localization changes between membrane-bound organelles. This would require much more detailed analysis by applying a high-resolution spatial proteomics approach, but would also be much more meaningful.

19. The authors state that based on the data shown in Figure 3B it can be seen that GRB2, SHC1 and CBL are rapidly reduced in their cytosolic localization (2 min EGF stimulation) and are recruited to the membrane fraction. If this is happening, one should also see an increase in intensity in fraction 3/4. Indeed, this can be seen for SHC1 and GRB2 based on the presented data, but not for CBL. The described shuttling back to the cytosol after 20 min is not reflected in the data for SHC1 and GRB2. It is unclear what happens with CBL given that there was basically no increase in fraction 4 after 2 min EGF stimulation but then again after 20 min a strong increase in the cytosolic signal is detected. Furthermore, it might be worthwhile to additionally measure total cell lysates following EGF stimulation as a reference to better see how much of the protein (in %) is localized to cytosolic (fraction 1/2) and membrane compartments (fraction 3/4).

20. It seems that for SHC1 the same Figure is shown in 3B and 3D. It is suggested to avoid redundancy in data presentation.

21. Line 231: The increase in SHC1 Y427 phosphorylation is equally seen in the cytosolic compartment (Fig. 3D). How do the authors explain this?

22. Line 232: Should be EGFR-mediated phosphorylation of SHC1.

23. Lines 253-274: The authors state that the EGFR phosphorylation sites show a dual distribution in liver, which is not observed in HeLa cells (Fig. 3G), and that this difference is in line with the distribution of endosomal marker proteins (Suppl Fig. 4C). However, the authors do not say whether this observation is of biological significance. The authors have previously shown the protein profiles in HeLa cells in comparison to kidney and liver samples (Suppl Fig. 4A and 4B). They observe differences in the profiles for lysosomal, extracellular matrix proteins as well as mitochondria. This difference in profiles is actually also seen for peroxisomal marker proteins. For mitochondria, they claim this difference in profiles might reflect morphological difference of mitochondria between these tissues (lines 256-257). However, as can be seen in Suppl Fig. 4A, the intensities of mitochondrial proteins in liver samples are shifted to fractions 1/2, which is also seen in the kidney samples although less pronounced. Based on these findings the authors must check the intactness of the organelles after tissue homogenization (e.g. by EM analysis). Differential centrifugation and density gradient profiles for mitochondria isolations from cell lines and different tissues are well established and have been successfully used for studies of the mitochondrial proteome (e.g. Pagliarini et al., 2008; doi: 10.1016/j.cell.2008.06.016). Such methodology can therefore be used to thoroughly examine the data presented here to reveal potential technical issues. Thus, in the reviewer's point of view the

current claims made by the authors are not justified and require a critical assessment of the obtained data on these tissue samples.

24. As to the dataset presented in Fig. 3A-3D, it would be interesting to see not only the benchmarking with EGFR but to know whether the authors were able to obtain any interesting new biological insights from this dataset?

25. Line 295: To which compartments the authors refer here?

26. For microscopy data in Fig. 4, a control without sorbitol should be shown. Also a quantification of the images should be performed. The legend describing the images shown in Suppl. Fig. 5D appears to be a copy of Fig. 4E and should be corrected.

27. In Figure 5B it is not clear what is significant here – is this decrease (fraction 2) and increase (fraction 6) really justified to claim it is significant? This would need validation. As stated also by the authors proteins of both ribosomal subunits show the same trend which is different to the observation made in the previous dataset (Fig. 4). Altogether, the data on ribosomal protein relocation and ribotoxicity appear to be preliminary. Osmotic stress and mechanical stress by muscle contraction are also quite different and it remains unclear why the authors tried to “validate whether the translocation of the ribosomal particles observed in vitro was also recapitulated in vivo after mechanical activation of the muscle.” (lines 359-361). It should also be noted that U2OS cells are not a model cell line for skeletal muscle with its distinct sarcomeric architecture. Also the two experimental conditions in terms of treatment appear to be not suited for a validation experiment, despite the fact that the observations were also different.

Minor Comments

1. It is suggested to remove “in-vivo” from the title as a major part of the work is performed in cell lines.

2. In the abstract, “...spatio-temporal regulation of cells,...” should rather be e.g. “spatio-temporal regulation of protein networks in cells,...”.

3. In the figure legend to Fig. 1 E, ‘protein’ clustering should better be named ‘phospho-site’ clustering.

4. It is not clear what parameters were chosen for the agglomerative cluster analysis. How were the number of cluster chosen for KMeans Clustering prior to MetaAnalysis? Maybe it would be beneficial to indicate the cluster assignment as a color code at the side of the heatmaps. It is not clear why both datasets have to be clustered together. Would it not suffice to cluster each separately and perform MetaMass Analysis on the results?

5. The F-Score in Fig. 2C is calculated on which samples exactly (HeLa, U2OS)?

6. According to the figure legend in Suppl. Fig 2E kinase intensities are shown, but given the high number of data it more likely refers to protein intensities. Please check.

7. For Suppl. Data 1, sheets and column headers should be described.
8. Suppl. Table 1 is very hard to read. I suggest to move row labels to the left.
9. In Fig. 3A, introduce PCG.
10. In Fig. 5B the exact p value should be written. I suggest, to mention the effect size in the text.

Reviewer #3 (Remarks to the Author):

Existing workflows in spatial proteomics are low throughput both in terms of biochemical fractionation, and in terms of MS analysis. Here, the authors have developed a workflow for higher throughput spatial proteomics, including the analysis of phosphopeptides. Such throughput is necessary to generate replicates for rigorous statistical analysis, and to pursue temporal dynamics. The resulting workflow uses detergent-based fractionation, avoiding the use of ultracentrifugation common to other methods, and is inherently higher throughput. The authors also make use of their recently published directDIA workflow which couples several cutting-edge LC-MS technologies, starting with the Evosep liquid chromatography system. This is connected to a modern mass spectrometer with ion mobility interface, which they have previously shown to obtain relatively deep proteomes with short LC gradients. The resulting spatial proteomes were shown to have superior resolution to a widely used commercial chemical fractionation kit. Conversely, the resolution was shown to be lower than existing centrifugation-based methods and therefore less informative for those interested in organellar assignments. Importantly, the method was used to effectively monitor localization changes both *in vitro* and *in vivo*. Moreover, the simultaneous spatial phosphoproteome permit a deeper interrogation into the events underlying relocalization.

This manuscript represents a valuable contribution to the spatial proteomics field and should encourage more labs to obtain subcellular localization information, particularly in a comparative format. Moreover, there is a growing sense in the proteomics community at large, that workflows must be shortened to become more mainstream. The application of cutting-edge acquisition and data analysis in this work, where previously known biology is recapitulated, lend further credence to this higher throughput mass spectrometry strategy and will benefit the wider community independently of whether they choose to use this chemical fractionation method or not.

Below I outline revisions which should further strengthen the manuscript.

Major Revisions

Line 73-74: Such a spatial phosphoproteome has been published previously by the Mann lab, using mouse liver and protein correlation profiling. Kraemer et al., 2018 Dev Cell (PMID 30352176). This study should be referenced and compared to the current study.

Line 123-124: The resulting plot shows a much lower correlation than expected, this could be due to protein loss during wash steps used in the method. Protein quantification of all fractions and washes would be important to assess this possibility, together with a silver-stained SDS-PAGE or alternative visualization method.

Line 138-139: Given the continued coupling of fractions 1 with 2, and 3 with 4, it warrants testing whether reducing the fractionation method to 1+2, 3+4, 5 and 6, would suffice. This analysis could be performed in silico on the existing data to determine the feasibility of this.

Line 168: The authors note that fractionation resolution is highly reproducible between these two cell lines, citing supplementary figure 2E. This statement is largely supported by the data. However, there do appear to be two clusters of proteins that are predominantly in fraction 2 in HeLa cells, but fraction 1 in U2OS. No fractionation procedure is impervious to such deviations, but it may help users of this method to understand where this might come from, to mitigate against it. Could the authors determine if there is a size bias or other property to these protein clusters.

The use of MetaMass as a benchmark lends credibility on the one hand, because it was published several years prior to this work, nonetheless, the scoring does appear lenient, which is manifested in two ways. The first is that the F-scores for other datasets are almost all close to 1, the second is that it is clear from plots shown in figure 1F, that membrane bound organelles are not well-resolved, yet the F scores would still imply significant predictive power, as detailed in lines 184-186. The authors should benchmark their method against these methods using an orthogonal metric, for example the QSep function published by the Lilley lab, described in Geladaki et al 2019. In addition, the previous spatial phosphoproteome obtained by protein correlation profiling in the Mann lab should be included in the MetaMass analysis (Kraemer et al 2018), as should work from the Borner lab cited in the introduction.

Line 258-274: The authors very elegantly identify a reason that phosphorylated EGFR is observed in the cytosol. However, why the lysosomes are found in the cytosolic fraction, when applying the method to tissues, is not discussed. The difference between the protocol for cultured cells and tissues is the use of the homogenizer, which may disrupt cell integrity. It would be valuable confirm this hypothesis experimentally, or otherwise determine the reason for lysosomes being found in the

cytosolic fraction. For example, to monitor cellular integrity, please check the post-homogenization supernatant for cytosolic markers.

Minor Revisions

Line 115: 7957 phosphorylation sites were identified from chemically fractionated HeLa cells, which is substantially less than 20,000 sites from unfractionated HeLa cells in the authors' recent publication. One would anticipate that fractionation would serve to increase depth. Please comment on this discrepancy.

Line 115-116, together with lines 620-622: 6952 proteins were quantified in total, yet only 4000 proteins per fraction implies a lot of imputation is required, yet the imputation process is insufficiently described.

Line 164: the word 'some' is interpreted liberally, it would appear that only three kinase substrate relationships have been analyzed. The authors should conduct a more systematic analysis or remove this statement.

Line 196: Use of the word comprehensive seems inappropriate, since the depth, although very good for short gradients, still falls short of comprehensive, where data from the Olsen lab serves as the current benchmark.

Line 218: Translocation of EGFR from the plasma membrane to early endosomes was observed in spatial proteomic studies from the Borner lab, this should be mentioned to highlight the information lost in this approach relative to centrifugation-based approaches.

Line 219: The authors state that they can 'clearly detect' how adaptor proteins reduce their cytosolic presence, and although the three proteins cited are in the top 2% (80/3883), the BH corrected p-value for SHC1 is higher than the routinely used 5% cutoff. This targeted assessment seems biased by prior knowledge, please adjust wording accordingly, or provide further details as to how hits were triaged.

Line 229-233: The authors infer a causal relationship which is not supported by this data alone but based on prior knowledge. EGFR phosphorylation is observed to increase, which is known to provide a docking site for SHC1. SHC1 is observed to increase membrane association, SHC1_Y427 phosphorylation is observed to increase in both the cytosolic and membrane fractions. The

conclusion that EGFR phosphorylation of SHC1 is a direct consequence of subcellular translocation cannot be deduced from these observations alone, because the simultaneous increase in the cytosolic pool of SHC1_Y427 could indicate alternative an alternative mechanism.

Line 276: replace 'proved' with the word 'demonstrated'

Line 346: Unclear where the hypothesis that RNA maturation was responsible comes from. Could the authors please elaborate on this.

Line 382: See comment above for line 73-74.

Line 437-438: Not clear how morphology and phenotypic differences will affect detergent performance, could the authors clarify this point.

Line 588: Why are MS1 scans collected until 1400 m/z if fragmentation is only performed on ions up to 1033 m/z?

Lines 589-592: Please replace Da with m/z.

Line 613: How or why are there protein identifications without valid gene names for the Uniprot database? If this is the case, could the protein names have been used or other identifier.

Point-by-point rebuttal letter

NCOMMS-21-09890-T:

“Spatial-proteomics reveals phospho-signaling dynamics at subcellular resolution” by Martinez-Val et al.

Our answers to the reviewer questions are indicated in blue text.

Reviewer #1 (Remarks to the Author):

We wish to thank the reviewer for constructive criticism and comments that we believe have improved our manuscript significantly. In particular, we have specifically addressed the parts concerning accumulation (increase) of the large subunit (LSU) of ribosomal proteins observed in the nucleolar fraction and a decrease in the cytosolic fraction observed in our proteomics data by providing new experimental validation data. Please find answers to the specific points raised by this reviewer below.

In the manuscript entitled “Spatial-proteomics reveal in-vivo phosphor-signaling dynamics at subcellular resolution” Martinez et al claim to develop a high-throughput workflow to rapidly fractionate and profile the proteome/phosphor-proteome at subcellular resolution. Using this workflow, they to investigate spatio-temporal proteomic/phospho-proteomic response during EGFR-phospho-signaling and osmotic stress at the cellular level, as well as during muscle contraction with the mouse model.

By sequentially treating cellular lysates to different detergents and different salt concentrations, the authors claim to chemically fractionate the subcellular contents. The contents of the six fractions were subjected to high-resolution mass spectrometry. The authors compare the performance of their workflow with a commercial kit used in a previous study (Mendes et al. 2017). As judged by mass spectrometry analyses, the protocol used in the current study seems to perform as well as the workflow reported by Mendes et al. 2017. This part of the the manuscript has been carefully performed.

We thank the reviewer for highlighting that our protocol performs at least on par with the commercial kit of secret composition due to proprietary information, and that our benchmark was carefully performed.

In my opinion, the biological (and mechanistic) insight(s) gained through the “less cumbersome” fractionation coupled mass spectrometry workflow to investigate EGFR-phospho-signaling, and in particular the causal link between osmotic stress/muscle stimulation and ribosomal protein accumulation seems limited.

By and large, we agree that the new biological and the mechanistic insights gained by the EGFR-phospho-signaling experiment and the accumulation of LSU proteins in the nucleus/nucleolus remain to a large extent descriptive. The manuscript as presented was intended to show an implementation of a straightforward and high-throughput approach to study spatio-temporal regulation of the proteome, in order to make this technical workflow

available to the scientific community. Both biological applications shown in the paper were intended as “proof-of-concept” experiments to validate the usefulness of the workflow. Firstly, EGFR signalling has been previously used in this field to evaluate spatial proteomics methods to study protein translocation. We used this model as a validation experiment to show that our workflow was capable of tracking EGF-mediated internalization and signaling. However, to address the concerns regarding the limited biological insights, we have in the new version of the manuscript devised and introduced a statistical approach to identify protein translocations in an unbiased manner from quantitative spatio-temporal proteomics data (see figure below and lines 267-276, Figures 4B and Suppl. Fig. 7A in the manuscript). Analyzing our EGFR signaling data with this new approach, does not only identify known EGF-dependent cytoplasmic-to-membrane translocation proteins such as Grb2, Shc1 and Cbl, but it also presents proteins previously unknown to translocate as a function of EGF stimulation.

Moreover, the osmotic stress application was originally also intended as a proof of principle experiment, but in a biological context that had not been explored with spatial-proteomics yet. Therefore, we used it to show the potential of the workflow to provide new data-driven hypotheses, as well as, how to further validate these findings. According to the reviewer’s

suggestion, we have now extended and improved the validation further as you will see in our answers below to the specific questions about the osmotic shock induced ribosomal protein accumulation. In summary, we think that an even more in-depth investigation of these new biological insights is beyond the scope of the manuscript, as the main novelty lies in the fractionation and the spatio-temporal proteomics and phospho-proteomics analyses.

Using cell biological studies, the authors claim an accumulation of the large ribosomal subunit proteins Rpl10a and Rpl22, but not small subunit ribosomal proteins, in the nucleus/nucleolus upon osmotic stress. The data presented are of poor quality and are not of the standard of the ribosome cell biology field. It would be important for the authors to show clear co-localization with additional nucleolar/nuclear markers.

We thank the reviewer for pointing this out and agree that the original validation data of Rpl10a and Rpl22 accumulation in nucleus/nucleolus upon osmotic stress, generally cannot match those presented in the ribosome cell biology field. Consequently, we followed the advice of the reviewer and performed new co-localization experiments with a specific nucleolar marker. Our MS-based proteomics data (Fig. 6B) showed a clear accumulation (increase) of Large Ribosomal Subunits (LSU) proteins in the nucleus/nucleolus (fraction 6) and a clear decrease in the cytosol (fraction 2) in response to sorbitol treatment. In contrast, we do not observe any significant changes in Small Ribosomal Subunits (SSU) proteins (Fig. 6B) indicating that the stress induced by the sorbitol treatment affects LSU proteins only. The immunostainings of Rpl10A, Rpl22 and Rps3 (Suppl. Fig. 10A-C) served as independent validation of the hypothesis derived from the proteomic data. To address the concern about quality and increase the quality of the follow-up experiments, we have included the quantification of the stainings (Suppl. Fig 10B and Panel B/C below) and included additional immunostainings of ribosomal markers (Rpl22 and Rpl10A) showing their co-localization with a specific nucleolar marker (Fibrillarin) (New Fig. 6E and Panel A below). With all of these improvements and new experiments, we hope that it is now convincing that the accumulation is specific to LSU proteins, and that the accumulation is seemingly in the nucleolus.

The nucleolar/nuclear accumulation of ribosomal proteins Rpl10a and Rpl22 was used as an argument for stalling of 60S pre-ribosomal particles in the nucleolus. If this were true, pre-rRNA levels should have increased in response to osmotic stress. Instead, Northern analyses indicate reduced pre-rRNA levels upon osmotic stress. If pre-rRNA processing was affected at least one of the pre-rRNA species should have accumulated. Rpl10a is incorporated into the pre-ribosome only in the cytoplasm, therefore its nucleolar/nuclear location is puzzling. The position of Rpl10a on pre-ribosomes is typically occupied by assembly factors/export adaptors, therefore it seems likely that free Rpl10a mislocalizes and accumulates in the nucleus/nucleolus. The authors need to address the issue whether Rpl22 and Rpl10a are bound to pre-ribosomes, to argue the point that 60S pre-ribosomal particles are accumulating in the nucleolus. The functional importance of accumulation of either of the ribosomal protein and/or pre-ribosome in the nucleolus upon osmotic stress and muscle contraction is descriptive. Overall, the study does not provide a fresh perspective to the links between critical anabolic processes and osmotic/muscle stress.

We apologize for the lack of clarity when interpreting and describing the results in the previous version of this manuscript. To solve this, we have extensively modified the results section describing these results to clarify that the main observation is the accumulation of LSU

proteins in the nucleolar fraction and the corresponding depletion in the cytosolic fraction, and not of their SSU counterparts . We have changed the phrasing from 60S ribosomal proteins to LSU proteins to emphasize that we only talk about the proteins and not the 60S (pre)-ribosomal particle as such, since there is no evidence to what extent the LSU proteins are accumulating together with a specific (pre)-ribosomal particle or as free proteins, as correctly pointed out by this reviewer (lines 358-448). We agree with the doubts raised by this reviewer regarding the northern blotting results of ribosomal RNA processing intermediates, and we believe that this is because the results were not comprehensively described. Thus, we have now extended this section to emphasize that these results serve to make the first attempt of linking the accumulation of LSU proteins to ribosomal RNA processing and to strengthen the observation that only LSU proteins are accumulating, which is in agreement with the northern blotting experiment. In summary, we see a strong decrease of 12S rRNA (Fig 6F and Suppl. Fig 11B-C) accompanied with an increase of the upstream processing intermediate, 32S rRNA (Fig 6F and Suppl. Fig 11B-C), together with an increased smear below the 32S rRNA band indicating that (some of) the accumulated 32S rRNA might get degraded (Fig 6F and Suppl. Fig 11C). We have also included a description of the overall decrease of processing intermediates in the sorbitol treated samples, which indicate an overall reduction in rRNA synthesis (lines 431-448). One potential hypothesis that we mentioned in the text (lines 427-428 and lines 544-546) is that this could likely be related to the displacement of nucleolin from the innermost fibrillar core, where it serves to stimulate rRNA synthesis among other functions, which is in agreement with published data (PMID: 18299322). Other potential mechanisms have been described in literature that links rRNA processing to ribosomal protein maturation and export. For instance, depletion of factors such as Plp7p, Sda1p, Rix1p or Rli1p proteins are connected to the defective synthesis of 5.8S rRNA, which leads to nuclear accumulation of pre-60S particles (PMID: 17509569).

We sincerely hope that this part of the manuscript is clearer and we would again like to thank this reviewer for the constructive criticism.

Reviewer #2 (Remarks to the Author):

Martinez-Val and coworkers developed a “high-throughput” mass spectrometry(MS)-based method for analyzing changes in the subcellular localization of proteins and phosphoproteins to study dynamic signaling events at a subcellular resolution. The described workflow for sample generation and LC-MS/MS analysis is used for temporal studies and the tracking of dynamic changes in the (phospho)proteome at a subcellular level. Analyses of EGFR signaling dynamics in HeLa cells as well as hypertonic stress-induced protein translocation in U2OS cells serve as proof-of-concept. Protein translocation by mechanical contraction is also tracked in mouse samples and some data are presented indicating relocation of ribosomal proteins.

What is notable about this work is that it aims at determining subcellular changes in the proteome and phosphoproteome in response to a stimulus or stress condition. It therefore advances spatial proteomics to the level of the phosphoproteome. For this, the authors apply a streamlined sample preparation workflow and state-of-the art MS methodology to reduce MS-acquisition time but still cover a large fraction of the proteome and phosphoproteome.

However, the chemical fractionation method they employ does limit their approach because the resolution it provides for discriminating between discrete changes in the subcellular localization of proteins and phosphoproteins is rather low. In fact, the authors also state that “this analysis revealed three major cellular compartments” when they discuss Fig. 1F. Based on the presented data (1F, Fig 2, Fig 3B etc.), this fractionation method is applicable to follow changes in protein localization between cytosolic compartments, membrane-bound organelles and nucleus/nucleolus but not between the different organelles, e.g. the ER and mitochondria (at least not at a moderate to high confidence level). The authors should present and discuss this limitation more clearly and consistently in their manuscript and, importantly, this limitation remains true regardless of the improvement of the chemical fractionation method compared to a commercial kit (Mendes et al.; Fig. 2) or use of high-end MS methodology.

We thank the reviewer for highlighting how our presented work advances the study of spatio-temporal proteomics to the phosphoproteome level. We agree that our workflow, despite its advantages in terms of increased throughput, is a little limited in terms of subcellular resolution. We have made this limitation more clearly in the text (lines 247-249).

One can also have different views on “high throughput” versus “resolution”. In the reviewer’s point of view this study would benefit from high(er) resolution at a subcellular level and a detailed analysis rather than performing a high number of different “proof-of-concept” studies with limited resolution and each with less informative value. Likely due to the rather high number of different analyses the authors present in this work, the overall presentation, analysis and interpretation of the acquired datasets is partly insufficient in detail and limited in depth and thoroughness.

In the literature, there is also increasing evidence that 30-50% of all proteins have dual or multiple subcellular localizations, a circumstance that is not mentioned and addressed in this work. The authors therefore might consider and assess how this is reflected in their spatial proteomics data and how does it affect data interpretation.

We thank the reviewer for pointing out the well-described fact that a significant fraction of the proteome shows multiple locations and for the suggestion to assess how this is reflected in our spatial proteomics data. Reassuringly, our data does not contradict this fact but supports it. We identify an average of ~5,000 proteins per fraction, that in total comprise ~7,000 proteins. This indicates that a high fraction of proteins is detected across multiple fractions. Indeed, only 17% of proteins were found in a single fraction, whereas 82% were found in two or more fractions, 1615 of which proteins are found in all fractions. However, it is important to highlight that our data does not state that protein location is static or limited to one compartment, but rather we assign protein location as the one where it is more abundant based on the protein profile across fractions as in previous publications (Itzhak et al - PMID: 27278775, Kraemer et al - PMID: 30352176).

To follow the reviewer’s advice, we have included a sentence (lines 133-136) to highlight that in our dataset 82% of proteins were found in two or more fractions and 23% of the proteins were transversely identified in all subcellular fractions.

Furthermore, to minimize the risk of overinterpretation, it is also suggested that the authors consistently apply statistical testing to each dataset and report the exact p-values to reveal

whether there is a significant change in the localization of a (phospho)protein based on the acquired data or whether it is just a tendency. Independent validation of data is suggested to support claims.

We appreciate that the reviewer points out the importance of applying statistical testing to validate the changes shown throughout the datasets. We already included that information in the first version of the manuscript (Supplementary Tables S6, S7, S11 and S12), where we performed limma (moderated t-test with Benjamini-Hochberg FDR correction) to assess changes after EGF treatment and sorbitol stress in all datasets (both in vivo and in vitro). In the new version of the manuscript, we have further indicated the specific p-values when needed.

Importantly, the suggestion from the reviewer inspired us to devise a new statistical method for unbiased analysis of protein translocation from quantitative spatial proteomics data termed “Movement Score” (lines 756-776), where we plot the magnitude of translocation between fractions against the statistical significance of the change. In this way, we can differentiate between insignificant trends and significant changes in protein translocation. We describe this analysis in the new version of the manuscript and include a new Figure 4B (see below) to visualize the significantly translocating proteins as a function of EGF stimulation.

We have included a description of this approach in the Method section “Identification of translocation events” (lines 765-776 and Supplementary Figure 12, see also below). Briefly, we have combined the statistical significance of the protein level change across time (derived

from the limma moderated t-test adjusted p-value for each fraction between time points) with the change of the protein distribution between compartments (termed as “Movement Score” or “MS”). Then we plotted both values, and filtered those proteins with a combined adjusted p-value < 0.05 and a MS > 10%.

First, we calculated the “Movement Score” to rank the proteins according to how much their profiles change between different time points (Panel #1 from figure below). To do so, we scaled the protein levels at a given time-point to the total abundance. Then we calculated the absolute difference between fractions at each time point against the control/initial condition, which represents the percentage of the protein that changes distribution. We selected the two compartments that showed the highest difference, since those would be the ones between which the protein likely moves. Then, we classify the potential translocation events in four categories: cytosol-nuclear (blue), cytosol-membrane (green), membrane-nuclear (red) and within the same neighborhood (grey), to make the visual inspection of the plots easier.

Secondly, based on the fact that proteins move between two compartments, we expect that to confidently identify a “moving” protein it need to change in abundance in both compartments (i.e., if a protein moves from the cytosol to the membrane, its intensity should decrease in the cytosolic compartment and increase in the membrane one). Therefore, we combined the p-values from the two relevant compartments using the Fisher’s method, followed by correction for multiple testing by Benjamini-Hochberg (Panel #2 figure below). Reassuringly, known EGF signaling proteins such as SHC1 surpass the significance threshold of 5%, since the p-value for cytosolic compartment movement (at 8 minutes) is 3.038932e-04 and the p-value for the membrane-bound compartment (at 8 minutes) is 0.0190969950. Consequently, using Fisher’s method, the combined p-value adjusted by BH results in 0.009.

Whilst the movement score evaluates the change in spatial distribution (i.e., translocation between cellular compartments), the combined p-value takes into account the statistical significance of that change. In previous subcellular proteomics papers translocation events have been assessed either by evaluating changes on the protein distribution profile using correlation metrics (Orre et al, Mol Cell, 2019, PMID: 30609389) without further statistical validation, or by elaborated outlier detection algorithms (Itzhak et al, eLife, 2016, PMID: 27278775). In contrast, here we present a straightforward statistical approach to estimate protein movement that can be easily applied to identify protein changes in time and space. Using this approach, we can see that GRB2, SHC1 and CBL appear as significantly translocating proteins at 2 and 8 minutes, and disappear at 20 minutes, matching the expected transient behavior on EGFR internalization, and with each translocation event validated by its corresponding adjusted combined p-value.

#1. Mobility Score

#2. Combined p-value

#3. Translocation Plots

To assess and validate our movement score, we took advantage of the fact that the EGFR signaling network is well-described and that we identify significant translocation of known EGFR signaling adaptors such as SHC1, GRB2 and CBL, which is in concordance with the literature (PMID: 30544639, 1322798, 23846654). We therefore do not consider it necessary to perform additional independent validation experiments.

That said, we agree with the reviewer that other statements from our work benefit from further independent validation, for example, the ribosomal translocation into the nucleolus observed after osmotic shock. Consequently, to independently validate our novel claims on ribosome

subcellular re-localization, we have included new co-localization experiments using immunofluorescence confirming our hypothesis of ribosome stalling on the nucleoli.

More specific suggestions that hopefully help the authors to improve their work are listed in the following:

Comments

1. Lines 48-51: Sentences with references 6-8. It is suggested to add further references to better represent published work.

We have included more references.

2. Line 59: “liquid chromatography coupled to mass spectrometry (LC-MS/MS)” should be “liquid chromatography coupled to tandem mass spectrometry (LC-MS/MS)”

We have corrected it.

3. The authors just refer to “Methods” when introducing the chemical fractionation method in the result part (Line 93). However, in the Discussion they describe and discuss this fractionation method and the different reagents used in detail. For better clarity, it is suggested to already introduce in the Results the properties of the fractionation chemicals and provide more details why these were used, what they do and what has been changed in the protocol compared to previous work.

In agreement with the reviewer, we have moved the description of the different buffer composition to the Result section (lines 95 to 107). We have also accompanied the description with Transmission Electron Microscopy Images of the different steps of the subcellular fractionation protocol, so it can be observed the effects of each one of the treatments in the cell (Figure 1B, and below).

4. The authors should provide additional information on the MS-acquisition method as the statement “more sensitive MS-acquisition method” (line 113) is not very informative. In addition, abbreviations used for LC-MSMS methods (SPD, CV) in Fig. 1A should be described in the Figure legend. Since the cell lines HeLa and U2OS are shown in the workflow, this should be described in the legend, clarifying for which cell line data are shown in Fig 1B-F. “LC-MSMS” should also be “LC-MS/MS”.

We have updated the Figure 1 legend to include the information requested, as well as described it in the text (lines 126-129).

5. Lines 117-121: This interpretation is not conclusive because in Fig. 1F it can be seen that nearly all marker protein profiles have low intensities in fraction 3. So, there is no subcellular compartment enriched in fraction 3.

We thank the reviewer for pointing this out and we understand the concern raised. It is true that protein profiles show lower intensities overall in fraction 3. In fact, it is the fraction that showed lower protein recovery (see our response below to the second comment of reviewer #3). However, we do not agree that no subcellular compartment is enriched in fraction 3. It is clear from the heatmap in Figure 2C that there is a significant number of proteins specifically purified in fraction 3. To further investigate the subcellular compartments purified in fraction 3, we performed Gene Ontology (GO) Enrichment analysis of the enriched proteins. This revealed that they belong to the lumen of several organelles such as endoplasmic reticulum (q-val: 1.06e-59), lysosomal lumen (q-val: 1.38e-17) or mitochondrial intermembrane space (q-val: 4.9e-9). Most importantly, this was also discovered thanks to another comment from this reviewer (see comment 7). In summary, we can conclude that fraction 3 is enriched in proteins from the lumen of membrane-organelles. We have added this information in the text in lines 167-176 and Supplementary Figures 1C, 2A and 2B.

6. It is not clear how the data was normalized. Was loess normalization applied to the complete dataframe? If so, wouldn't the assumption that the majority of proteins between samples is unaltered be violated? It would be nice if the material and methods would suffice to recapitulate the data processing. Also, it is not clear how the intensities for the profile plots in Fig. 1F were scaled.

We apologize for the confusion; we did the normalization separately for each fraction such that we do not alter the relative intensity of proteins between fractions. In summary, for our data analysis we treated each fraction independently throughout the whole data analysis process. Normalization and missing value imputation was performed independently for each fraction, and only afterwards, the fractions were merged together, but after this step no further normalization was performed to keep the relative intensity across fractions. We mentioned this in the Method section “Data analysis” (lines 732-734): “For normalization, to remove experimental bias, as well as for imputation of missing values, each fraction was treated separately.” To clarify this better, we have extended the data analysis description to explain this process in more detail. Also, the R-code used for processing will be made available in the GitHub project page.

Concerning the scaling of intensities in profiles plots (such as in Fig 2F), once the data per fraction was normalized and imputed, we merged the information for all six fractions. Missing values resulting after merging the six fractions (i.e.: proteins not identified in one of the

fractions) were imputed using a left-censored distribution. Afterwards, we transformed the values from log₂ intensities to raw values and calculated the sum of total protein intensity across fractions (per replicate). Finally, for each replicate, we divided the intensity of each protein per fraction by the summed intensity of the protein in the experiment. Therefore, the scaled intensity represents the relative value (from 1 to 0) of the abundance of that protein across fractions.

7. The authors should comment why lysosomes and ER marker proteins have two distinct peaks in the average profile plots (Fig. 1F) and how many of these follow one of the two different profiles – what is the difference between the two populations? Have the different marker protein lists been checked manually for correctness and proteins with multiple localizations?

We thank the reviewer for this interesting observation and raising the question about why ER and lysosome markers follow a dual distribution between fractions 3 and 4. The marker lists used were extracted from the proLoc R package (Gatto et al, 2014, PMID 24413670), which are used as canonical subcellular location markers to predict subcellular location. First, as suggested by the reviewer, we further validated those markers by performing Gene ontology enrichment in StringDB (v11). 73 protein markers were used to define the ER compartments. All of them are annotated within the Endoplasmic Reticulum GO-CC term with a q-value of 1.41e-73. Therefore, we consider these markers as ground truth representatives of the corresponding compartments.

Considering this, our data shows that ER proteins are purified in two different fractions: out of 73 proteins, 22 are purified in fraction 3 and 51 in fraction 4 (see figure below, Panel A). We performed further analysis to identify an explanation for this separation. We performed GO analysis on each subset in StringDB (v11) and found out that ER proteins purified in fraction 3 are annotated as luminal proteins (14 out of 22, q-val:6.5e-19), whilst ER proteins purified in fraction 4 were bound to the ER membrane (48 out of 51, q-val: 2.4e-55). Same trend was observed in lysosomal markers, where those annotated as present in the lysosomal lumen predominantly purified in fraction 3.

However, since the lysosomal marker dataset consisted of only 25 proteins, the GO enrichment analysis was not discriminating (Panel B). Moreover, the dual distribution between fractions 3 and 4 was not as consistent as for ER proteins, as shown by the reproducibility between replicates. To better interpret the dual distribution of lumen/membrane proteins in fractions 3 and 4 of our dataset, we mapped all proteins annotated in GO-CC terms as “lysosomal lumen” or “lysosomal membrane”. For that purpose, we downloaded the complete gene annotation of Homo sapiens from the Gene Ontology Consortium (2021-02-01 release), and filter by GO:0005765 (“Lysosomal membrane”) and GO:0043202 (“Lysosomal lumen”). The majority of proteins annotated as “Lysosomal membrane” were found in fraction 4 (Panel C). Interestingly, a small proportion of this last set of proteins were also found in fractions 1 and 2. However, manual inspection of these revealed that they were proteins such as MTOR and MTOR-associated proteins, which, although being annotated as “Lysosomal membrane” proteins, are not primarily membrane-bound proteins and they are also not found in the interior of the lysosomes, and consistently they are mainly quantified in the cytosolic fractions (FR1-FR2). Our findings suggest that the potential association of these proteins with the lysosomal membrane happens on the outer part of the vesicles, which is consistent with the literature

(PMC3840941). Importantly, this analysis indicates that using GO-terms to annotate subcellular location is limited and can lead to wrongful assignments, highlighting the relevance of subcellular proteomics to curate current annotations.

In fact, we predicted the proportion of transmembrane regions among these proteins (<http://www.cbs.dtu.dk/services/TMHMM/>) and we found out that proteins purified in fraction 4 had higher probability of having transmembrane regions than those found in other fractions (Panel D). Therefore, confirming that membrane-bound proteins from the lysosome are purified in fraction 4. Conversely, proteins associated with “Lysosome lumen” were distributed in either fraction. However, none of those proteins had a significant proportion of transmembrane regions (average number of transmembrane helices = 0.25), which indicates that for lysosomal cargo proteins, the purification was not specific to fraction 3, as observed for the Endoplasmic Reticulum.

In summary, this new analysis indicates that the chemical composition of buffers 3 and 4 determines the selective purification of soluble protein from within membrane-organelles (extracted with buffer 3 based on Tween 20), from those bound to the membranes (extracted with buffer 4 containing DDM).

We have included this new information in the manuscript in lines 166-176 and supplementary figures 2A and 2B.

8. The authors should comment on the relationship between Fig. 1D and Fig. S1B. If both show relationships between the fractions, how can it be that in the agglomerative clustering fraction 3 is closer to 5 and 6, while in Fig. S1B fraction 3 seems to have an overall higher correlation with fraction 4 and 1?

Figure 1D (now Figure 2C in the new manuscript version), as well as all clustering figures throughout the manuscript are performed using Euclidean distance for sample clustering, whilst Fig S1B is based on Pearson correlation. If we repeat the clustering of figure 1D with Pearson distance, fraction 3 is clustered closer to fraction 4.

In any case, the purpose of the clustering shown in figure 2C-D (and similar figures in the text) is to show that the separation between different fractions is much higher than the separation between replicates for the same fraction, rather than showing similarities between fractions.

As suggested by reviewer #2 in comment 16, we have now included a t-SNE analysis (Figure 2E and below) which is a better representation of the similarities and differences across different fractions.

9. Lines 122-127: The experiment described has been conducted in HeLa cells. U2OS cells are first introduced in line 166. Therefore, it is not apparent why the correlation of the summed intensities of the fractions and the intensities of the whole cell lysate of the U2OS cell line is shown in Fig. S1A. It is suggested to introduce both HeLa and U2OS when referring to Fig. 1. In addition, the intensities in both approaches should increase linearly and thus the Pearson correlation coefficient would be more appropriate to use.

Thanks for pointing this out, we have now included in that figure (now Supplementary Figure 3D and below) the experiment performed in both HeLa and U2OS cells to avoid confusion. Also, we have included the Pearson, as well as the Spearman correlation in the figure.

10. Lines 142-148: The authors refer to GO enrichment analysis and that it “revealed even more specific pattern for each fraction...” Unfortunately, these data are only presented in a Suppl. Table. GO term enrichment analysis would be indeed helpful to better judge the

resolution of the fractionation approach and thus it is recommended to include the data as a plot for each fraction as in Fig. 1F.

We agree with the reviewer's opinion and we have now included two supplementary figures to illustrate that data (Supplementary Figure 1C and 1D).

11. Line 148: Using the Cell Atlas data as reference cannot be considered as validation of the acquired data. It is also suggested to mention in each plot in Suppl Fig. 2A how many protein profiles are shown. This also applies to e.g. Suppl Fig. 2B.

We have now rephrased the sentence and added the number of protein profiles shown in each plot.

12. Lines 152-157: Please show that this enrichment in fraction 1 is significant. The authors should also clarify to which ribosomal proteins they specifically refer. The conclusion is too speculative.

We have observed that phosphorylated forms of RPS6 (tri-phosphorylated at sites S235, S236 and S240), RPL14 (S139), RPL24 (S86) are mostly purified in the first fraction, whilst the non-phosphorylated forms of those proteins are mainly purified in fraction 2. Interestingly, other phosphorylated forms of RPS6 show a dual distribution between fraction 1 and 2. We considered that this observation is relevant since there is extensive research on the differential function of ribosomal proteins (and of RPS6 in particular) when phosphorylated (REFs). For instance, in a recent preprint (doi: <https://doi.org/10.1101/2021.03.18.436059>) it is discussed that RPS6 phosphorylation is dynamic and depends on the ORF length. Moreover, the preprint also reports that RPS6 from ribosomes associated to the endoplasmic reticulum are more rapidly dephosphorylated than cytosolic ribosomes, suggesting a dependence between subcellular location and phosphorylation status. Nevertheless, we understand that at this stage and limited to the data presented here, any further affirmation is purely speculative, so we have rephrased it and toned it down in the main text (lines 207-210)

13. Lines 160-162: Statement is not justified; see comment 5.

We hope we address this concern in the response to comment 5. Now we have edited the manuscript to clarify this issue about proteins enriched in fraction 3 (lines 167-176).

14. Lines 162-165. The authors should comment why these kinases and substrates were selected including the source of annotation and why not a more unbiased data analysis has been performed? They should also mention the subcellular localization of each of these kinases.

The kinases were chosen based on previous knowledge extracted from the PhosphoSitePlus (PSP) webpage. Briefly, we selected well-known kinases covering the three main compartments observed with our workflow (cytosol, membranes and nucleus) and looked in PSP for known substrates of these kinases that were also observed in our dataset. Nevertheless, we understand the concerns of this reviewer about this biased analysis, and prompted by this, we decided to include a more extended and unbiased analysis of the kinome-substrate subcellular landscape. For that purpose, we annotated all kinases known to phosphorylate substrates identified in our dataset using PSP information with the Perseus

computational platform. Then we annotated for each kinase and each substrate phospho-site their main subcellular location and plotted the resulting information as a network (see Panel A below), where kinases are grouped based on their main location (indicated by the color of the outer ring of the node) and their interactions extracted from StringDB. Then we included the substrate information as a pie-chart inside each kinase, where all its known substrates observed in our data are plotted, again shown with a color-code indicating their main subcellular location. We selected some of the most promiscuous kinases in our dataset (those with many known substrates) to visualize in more detail their kinase spatial landscape: GSK3B, MAPKAPK2, AURKA, EGFR, PRKAA1. On the one hand, we found that the subcellular distribution of substrates for some kinases are restricted to their main location, such as EGFR or PRKAA1. However, for other kinases, we show that the location of their substrates does not necessarily match the location of the kinase itself, such as GSK3B or MAPKAPK2. This either suggests the existence of mobility of the kinase to exert its function, or, on the contrary, the mobility of the substrates once they are phosphorylated.

We have included this new information in the manuscript lines 218-228 and Supplementary Figure 5D-E.

15. Suppl Fig. 2E-G: It is suggested to present this comparison earlier in the Results part.

We have re-structured that section on the results to introduce the comparison earlier in the text.

16. The data obtained in this work are first compared to a published dataset using a commercial kit for subcellular fractionation, and compare favorably as judged by F-score statistics as shown in Fig. 2C. However, it is not easy to judge whether the sequential cell fractionation protocol provides a better partitioning compared to the previous approach as the analysis pipelines used greatly differ (label-based quantification, FAIMS, DDA, CID vs. label-free, DIA, HCD). Comparison of the described fractionation protocol against established protocols using the same analysis pipeline would be of interest. Furthermore, visualization of the data after dimension reduction (e.g. principal component analysis, tSNE etc.) is suggested to make the comparison more meaningful. The authors should also state that the LOPIT-based and SubCellBarCode methods clearly show better partitioning of subcellular localizations as visible from F-scores shown in Suppl. Fig. 3E.

We appreciate the comment of this reviewer and their suggestions. However, we do not want to claim in the current manuscript that our protocol provides better partitioning. In fact, in the metaclass-based comparison with other published dataset we show that our approach shows more limited resolution than differential centrifugation or density gradient approaches. Our strong point in favour of our protocol is how it can be applied in a high-throughput, not only thanks to a simpler fractionation approach, but also due to the automatized sample preparation in the KingFisher and the single-shot data acquisition using DIA, which reduces sample acquisition time and favors reproducibility. We consider that applying this pipeline to other subcellular fractionation approaches won't be fair in our hands, since we do not have a well-established protocol for either differential centrifugation or density gradient approaches, so any result would be biased by that. We agree that the comparison with other protocol using a similar pipeline would be of interest, but we do not consider that to be in the scope of our current manuscript.

However, we have stated clearly in the new version of the manuscript that LOPIT-based and SubCellBarCode methods show better partitioning of subcellular locations (lines 247-249).

Finally, following this reviewer suggestion we have performed tSNE analysis on our data (Figure 2E).

17. Lines 184-186: It is not clear what exactly was unexpected. Were the F-scores for the membrane bound compartments unexpectedly high given that these organelles were not well separated, or does this refer to the comparison to previous results?

We have revised the text: The F-scores for markers of the plasma membrane, endoplasmic reticulum, lysosomes and mitochondria were in the range of 0.69-0.85.

18. Lines 190-192: Why is the precision of centrifugation-based methods higher as expected? It is also not clear what the authors mean with "approximate locations"? The authors should be more precise here. Also from these data it is evident that the chemical fractionation approach is very good in localizing proteins to cytosolic compartments, membrane-bound organelles in general and nucleus/nucleolus but that it is considerably less

effective in revealing protein localization changes between membrane-bound organelles. This would require much more detailed analysis by applying a high-resolution spatial proteomics approach, but would also be much more meaningful.

We have revised the text to clarify. First, we underscore the point made by the reviewer: separation of intact organelles such as density centrifugation and differential centrifugation yields higher spatial resolution than what is obtained by chemical fractionation. However, since these methods yield fractions with very high protein overlap, the resolution comes at the cost of a high amount of instrument time. Chemical separation yields fractions with very low overlap, and the MS usage is significantly reduced when compared to other methods with more fractions, and hence higher resolution, but very MS-time consuming.

As the reviewer explains, the expectation is that chemical fractionation is limited to discriminate between cytosol, nucleus and membranes. We were therefore surprised to observe that we obtained F-scores in the range of 0.6-0.7 for the plasma membrane, ER and mitochondria. This results shows that there is some discriminatory power in the membrane compartment. In the discussion we explain how this may be increased further by implementation of cell surface biotinylation.

19. The authors state that based on the data shown in Figure 3B it can be seen that GRB2, SHC1 and CBL are rapidly reduced in their cytosolic localization (2 min EGF stimulation) and are recruited to the membrane fraction. If this is happening, one should also see an increase in intensity in fraction 3/4. Indeed, this can be seen for SHC1 and GRB2 based on the presented data, but not for CBL. The described shuttling back to the cytosol after 20 min is not reflected in the data for SHC1 and GRB2. It is unclear what happens with CBL given that there was basically no increase in fraction 4 after 2 min EGF stimulation but then again after 20 min a strong increase in the cytosolic signal is detected. Furthermore, it might be worthwhile to additionally measure total cell lysates following EGF stimulation as a reference to better see how much of the protein (in %) is localized to cytosolic (fraction 1/2) and membrane compartments (fraction 3/4).

We thank the reviewer for this observation and suggestion. To best evaluate translocation of proteins, we took into consideration the alteration of the distribution between fractions for each time point, which is how translocation has been defined in previous publications on this field (Itzhak et al, 2016 PMID: 27278775 and Orre et al, 2019 PMID: 30609389). Considering this, our data shows that in initial unstimulated conditions, CBL, SHC1 and GRB2 proteins are mostly detected in the cytosol. After EGF stimulation for 2 min, we observed that the distribution of these proteins changed, decreasing their relative cytosolic fraction and increasing the membrane one. This change in the distribution is swapped after 20 minutes of treatment (CBL and SHC1) and for 90min for GRB2, when the majority of the protein is again measured in the cytosolic fraction. Additionally, we have included a scoring approach to measure translocation of proteins. Using this scoring approach, we evaluated a translocation event based on the change in distribution between compartments and the statistical significance of that change (see Supplementary Figure 12). As a result, we observed that CBL, SHC1 and GRB2 are marked as translocating proteins at 2 and 8 minutes, but not after 20 minutes (see Figure 4B-C and Panel A below).

Additionally, following the reviewer's suggestion, we have measured the whole cell lysates of these same cells and time points to evaluate whether their overall abundance changes in response to EGF (Panel D from Figure below). From this analysis, we did not see any

significant change in protein levels for EGFR, CBL, SHC1 and GRB2 at 2, 8 and 20min. However, we observed a significant decrease of EGFR levels at 90 minutes (Fold Change: -1.65, q-value= 9.8e-5), which matches the expected degradation of this receptor in response to EGF stimulation (PMID: 27136326).

20. It seems that for SHC1 the same Figure is shown in 3B and 3D. It is suggested to avoid redundancy in data presentation.

We have re-designed the figures associated with that data (new Figure 4) to reduce this redundancy.

21. Line 231: The increase in SHC1 Y427 phosphorylation is equally seen in the cytosolic compartment (Fig. 3D). How do the authors explain this?

We have included a mention to this fact in the manuscript (lines 306-309). Whereas we present in the text that the phosphorylation of SHC1 observed in the membrane-enriched fraction might be connected to EGFR internalization due to EGF stimulation, it is very clear by our data that SHC1 is also phosphorylated in its cytosolic form. This fact might indicate that other

tyrosine kinases are active in the cytosol and therefore can phosphorylate the cytosolic SHC1 form. In fact, some cytosolic tyrosine kinases, such as SRC, are known to phosphorylate SHC1 and it has also been shown that SRC can trigger phosphorylation of its substrate upon EGFR activation (Begley et al 2015, PMID: 26551075).

22. Line 232: Should be EGFR-mediated phosphorylation of SHC1.

Thanks, this is now corrected.

23. Lines 253-274: The authors state that the EGFR phosphorylation sites show a dual distribution in liver, which is not observed in HeLa cells (Fig. 3G), and that this difference is in line with the distribution of endosomal marker proteins (Suppl Fig. 4C). However, the authors do not say whether this observation is of biological significance. The previously have shown the protein profiles in HeLa cells in comparison to kidney and liver samples (Suppl Fig. 4A and 4B). They observe differences in the profiles for lysosomal, extracellular matrix proteins as well as mitochondria. This difference in profiles is actually also seen for peroxisomal marker proteins. For mitochondria, they claim this difference in profiles might reflect morphological difference of mitochondria between these tissues (lines 256-257). However, as can be seen in Suppl Fig. 4A, the intensities of mitochondrial proteins in liver samples are shifted to fractions 1/2, which is also seen in the kidney samples although less pronounced. Based on these findings the authors must check the intactness of the organelles after tissue homogenization (e.g. by EM analysis). Differential centrifugation and density gradient profiles for mitochondria isolations from cell lines and different tissues are well established and have been successfully used for studies of the mitochondrial proteome (e.g. Pagliarini et al., 2008; doi: 10.1016/j.cell.2008.06.016). Such methodology can therefore be used to thoroughly examine the data presented here to reveal potential technical issues. Thus, in the reviewer's point of view the current claims made by the authors are not justified and require a critical assessment of the obtained data on these tissue samples.

We appreciate the comment of the reviewer and we understand the concerns raised. We are aware that the cell integrity of tissues is compromised in our current approach, since we perform homogenization of frozen samples. On the one hand, this approach allows us to preserve the in-vivo state of labile modifications such as phosphorylation, which otherwise will be subjected to enzymatic removal by phosphatases or chemical degradation, losing essential information for signaling studies such as the one presented in here. On the other hand, it impairs cellular integrity, resulting in worse organelle and subcellular compartment resolution. The most relevant consequence in this regard is how endosomal markers are co-purified in the first fraction, which is mainly enriched in soluble cytosolic proteins. To understand this issue better, we have performed electron microscopy imaging as suggested by the reviewer. By transmission electron microscopy (TEM), we show that despite the damage of macro-cellular structure, the intracellular interactions remain intact, as well as the intracellular organelles (i.e: mitochondria, Golgi, nucleus) allowing the chemical fractionation to work reproducibly. In the panels below you can see TEM images of different snapshots of liver tissue (A-E) and kidney tissue (F-J) at the different steps of the subcellular fractionation protocol. Panels A and F show the tissue just after the homogenization procedure. It is clear that the cell integrity is broken, however, the result from the homogenization is the form of multiple membrane-bound structures, which we hypothesized are remnants from the cell.

which should contain cytosolic proteins. Importantly, we can see that the nucleus remains intact (black arrows). In addition, we can observe the rest of the Golgi (light blue arrow) and intact mitochondria (red arrows). Through the following steps, we can still see how mitochondria remain intact to digitonin treatment and 0.140mM of salt (panels A/B and F/G, fractions 1 and 2, red arrows). At this step we can also clearly see the Golgi network (blue arrows). For fractions extracted with detergents, such as Tween-20 (panel D and I, fraction 3) and DDM (panel E and J, fraction 4), we can observe how the membrane-bound vesicle-like structures are dissolved and only the nucleus remains.

Another important observation that this reviewer highlights is the different profile observed for mitochondrial protein between HeLa, liver and kidney. As this reviewer points out, the intensities of mitochondrial proteins in liver samples are shifted to fractions 1/2, which is also seen in the kidney samples although less pronounced. In contrast, in HeLa the mitochondrial proteins are specifically found in fraction 4. We initially hypothesized that this could be due to morphological differences between tissues; however, we did not have data to back it up. However, thanks to TEM we could observe and monitor mitochondria through the subcellular fractionation protocol. TEM images show that mitochondria are swollen due to the hypotonic buffer in both liver and kidney samples, but they also show that, despite the loss of the cellular structure, mitochondrial integrity is preserved up to the detergent extraction step (fraction 4). Importantly, TEM reveals an important difference in the overall tissue integrity between liver and kidney that might explain the difference in mitochondrial profiles. Whilst liver tissue loses completely all macro-cellular ultrastructure, the kidney retains some of it. Therefore, whilst in the liver, mitochondria are exposed to the action of all buffers from the very beginning; we observed that they are contained within bigger structures in the kidney. This can explain how in the liver we observe mitochondrial protein very early, potentially due to the permeabilization of mitochondrial membrane due to the action of digitonin.

We have modified the manuscript accordingly to include this information (lines 332-339 and lines 513-538).

24. As to the dataset presented in Fig. 3A-3D, it would be interesting to see not only the benchmarking with EGFR but to know whether the authors were able to obtain any interesting new biological insights from this dataset?

Thanks to the new “Movement score” approach described above, we identify several potential new targets that relocate in response to EGFR stimulation. We have included some relevant examples in Figure 3D, which includes PRELID1, CHUK, RNF220 and ELL2, each one reflecting a different translocation event. Among those, we found CHUK to be of highest interest. CHUK (or inhibitor of nuclear factor kB kinase-alpha) has been described as located both in the nucleus and the cytosol, and shuttle between them (Birbach et al, 2002, PMID: 11801607), which confirms the initial distribution observed in our dataset. The crosstalk between CHUK and EGFR has been pointed out as highly relevant in some specific biological conditions, such as keratinocyte proliferation in squamous cell carcinoma (Liu et al, 2008, PMID: 18772111). Liu et al (Liu et al, 2008, PMID: 18772111) described upregulation of EGFR in IKKalpha null epidermis, and that EGFR inhibition in IKK-alpha cells blocked proliferation. Our data revealed that during the peak of EGF-stimulation (2 to 20 minutes), the nuclear fraction of CHUK decreases, and then it is recovered after 90 minutes (when EGFR signaling is shut down due to EGFR degradation). This suggests that CHUK-EGFR regulation might be mediated by changes in subcellular localization, and that disappearance of CHUK from the nucleus might be a regulatory mechanism to repress its function and promote cell growth and mitosis.

25. Line 295: To which compartments the authors refer here?

We refer to cytosol and nucleoli, corresponding to fraction 2 and 6 respectively.

26. For microscopy data in Fig. 4, a control without sorbitol should be shown. Also a quantification of the images should be performed. The legend describing the images shown in Suppl. Fig. 5D appears to be a copy of Fig. 4E and should be corrected.

We have included the control without sorbitol (Suppl. Fig 10C), as well as the quantification (Suppl. Fig. 10B). Also, the legend has been corrected.

27. In Figure 5B it is not clear what is significant here – is this decrease (fraction 2) and increase (fraction 6) really justified to claim it is significant? This would need validation. As stated also by the authors proteins of both ribosomal subunits show the same trend which is different to the observation made in the previous dataset (Fig. 4). Altogether, the data on ribosomal protein relocation and ribotoxicity appear to be preliminary. Osmotic stress and mechanical stress by muscle contraction are also quite different and it remains unclear why the authors tried to “validate whether the translocation of the ribosomal particles observed in vitro was also recapitulated in vivo after mechanical activation of the muscle.” (lines 359-361). It should also be noted that U2OS cells are not a model cell line for skeletal muscle with its distinct sarcomeric architecture. Also the two experimental conditions in terms of treatment appear to be not suited for a validation experiment, despite the fact that the observations were also different.

We agree with the reviewer that osmotic stress and muscle contraction-induced mechanical stress are quite different. Importantly, however, there are also commonalities that we believe are relevant to this study. Hyperosmotic shock and muscle contraction share many key

features including isotropic pressure to the cell/muscle fibers. Both of these treatments increase membrane pressure and mechanical strain to the cytoskeleton (Chen, 2008, PMID: 18843115). In search of a physiological process that could reasonably be expected to be accompanied by translocation of ribosome particles to the nucleolus, we thus decided to investigate muscle undergoing in situ contraction. As it was written in our original manuscript, it indeed appears as if we consider in situ muscle compression to be an in vivo counterpart of hyperosmotic shock. This was not intended, and we have now changed the text accordingly (lines 454-459).

Minor Comments

1. It is suggested to remove “in-vivo” from the title as a major part of the work is performed in cell lines.

We have modified the title accordingly.

2. In the abstract, “...spatio-temporal regulation of cells,...” should rather be e.g. “spatio-temporal regulation of protein networks in cells,...”.

We appreciate the suggestion, and we have included it in the abstract.

3. In the figure legend to Fig. 1 E, ‘protein’ clustering should better be named ‘phospho-site’ clustering.

We have corrected this in the legend.

4. It is not clear what parameters were chosen for the agglomerative cluster analysis. How were the number of cluster chosen for KMeans Clustering prior to MetaAnalysis? Maybe it would be beneficial to indicate the cluster assignment as a color code at the side of the heatmaps. It is not clear why both datasets have to be clustered together. Would it not suffice to cluster each separately and perform MetaMass Analysis on the results?

We used Euclidean Distance for K-means clustering. The number of clusters were selected to yield an average of five proteins per cluster. With fewer clusters, more of the dataset is classified while F-scores decrease (see below figure).

In the revised version of MetaMass II, users can add two stringency parameters: “Group size threshold” is used to exclude small clusters from the analysis. “Marker count threshold” is used to exclude clusters with few markers.

One can of course cluster datasets separately, but for the purpose of this study, it was more rational to merge datasets. Thus, datasets obtained in this study were merged and clustered as one file while datasets from the reference study were merged and clustered as a second file. The F-scores therefore refer to the merged datasets. The difference between F-scores obtained using different fractionation methods is largely the same if one analyses a single or a merged dataset.

5. The F-Score in Fig. 2C is calculated on which samples exactly (HeLa, U2OS)?

F-scores are calculated from the analysis of merged datasets, so in this case it was the merged datasets from HeLa and U2OS.

6. According to the figure legend in Suppl. Fig 2E kinase intensities are shown, but given the high number of data it more likely refers to protein intensities. Please check.

We thank the reviewer for noticing for this mistake, it was a typo in the legend. The heatmap corresponds to all proteins identified in HeLa and U2OS datasets, not only the kinases. This has been corrected in the new supplementary figures document, where that figure is now located in Supplementary Figure 3A.

7. For Suppl. Data 1, sheets and column headers should be described.

We have included a description of the content of each sheet in all supplementary data tables.

8. Suppl. Table 1 is very hard to read. I suggest to move row labels to the left.

We have edited this table to solve this issue.

9. In Fig. 3A, introduce PCG.

We have included the definition of PCGs in the legend on Fig 3A.

10. In Fig. 5B the exact p value should be written. I suggest, to mention the effect size in the text.

This figure has been corrected and the asterisks have been replaced by the exact p-values.

Reviewer #3 (Remarks to the Author):

Existing workflows in spatial proteomics are low throughput both in terms of biochemical fractionation, and in terms of MS analysis. Here, the authors have developed a workflow for higher throughput spatial proteomics, including the analysis of phosphopeptides. Such throughput is necessary to generate replicates for rigorous statistical analysis, and to pursue temporal dynamics. The resulting workflow uses detergent-based fractionation, avoiding the use of ultracentrifugation common to other methods, and is inherently higher throughput. The authors also make use of their recently published directDIA workflow which couples several cutting-edge LC-MS technologies, starting with the Evosep liquid chromatography system. This is connected to a modern mass spectrometer with ion mobility interface, which they have previously shown to obtain relatively deep proteomes with short LC gradients. The resulting spatial proteomes were shown to have superior resolution to a widely used commercial chemical fractionation kit. Conversely, the resolution was shown to be lower than existing centrifugation-based methods and therefore less informative for those interested in organellar assignments. Importantly, the method was used to effectively monitor localization changes both *in vitro* and *in vivo*. Moreover, the simultaneous spatial phosphoproteome permit a deeper interrogation into the events underlying relocalization.

This manuscript represents a valuable contribution to the spatial proteomics field and should encourage more labs to obtain subcellular localization information, particularly in a comparative format. Moreover, there is a growing sense in the proteomics community at large, that workflows must be shortened to become more mainstream. The application of cutting-edge acquisition and data analysis in this work, where previously known biology is recapitulated, lend further credence to this higher throughput mass spectrometry strategy and will benefit the wider community independently of whether they choose to use this chemical fractionation method or not.

Below I outline revisions which should further strengthen the manuscript.

We wish to thank the reviewer for the thorough assessment and positive comments on our manuscript, and for highlighting the valuable contribution this represents and the importance of the cutting-edge acquisition and data analysis described. We have followed the advice and recommendations made by the reviewer for revising the manuscript and we believe that this has improved our manuscript significantly.

Major Revisions

Line 73-74: Such a spatial phosphoproteome has been published previously by the Mann lab, using mouse liver and protein correlation profiling. Krahmer et al., 2018 Dev Cell (PMID 30352176). This study should be referenced and compared to the current study.

We thank the reviewer for pointing this out, and we have included the reference to this study in our new version of the manuscript (line 73).

We have also compared the phospho-profiles subcellular fractionation obtained by Krahmer et al using the F-score calculated by Metamass using subcellular markers obtained from Uniprot subcellular annotation. As it can be seen below (and in Suppl Fig 6F), the capacity to resolve the different compartments at phospho-proteomics level is highly similar between our

approach and Krahmer et al approach. Some relevant differences are observed in our liver phospho preparations in the lysosome F-scores. That is due to the presence of lysosomal proteins in the cytosolic fraction in our liver preparation, which we described in the text as a consequence of the rupture of the cell integrity during homogenization. However, overall, the F-scores obtained for each compartment in the phospho-peptide purifications are lower than those obtained for full proteome. Upon observation of the heatmaps showing the profiles at protein and phospho-protein level of Krahmer et al (Figures 2A and 2B, respectively from the corresponding paper), it seems clear that the fractionation provides much better defined profiles at protein than at phospho level in that study. This could be explained due to the fact that phosphorylated species of one protein can change location, or that markers from subcellular compartments are defined for the protein and there is not accurate annotation for the phosphorylated species. In this regard, that could also explain the poorer F-scores obtained at this level, since these scores are calculated based on annotation of location of proteins from full proteome studies.

Line 123-124: The resulting plot shows a much lower correlation than expected, this could be due to protein loss during wash steps used in the method. Protein quantification of all fractions and washes would be important to assess this possibility, together with a silver-stained SDS-PAGE or alternative visualization method.

We agree with the reviewer that some protein loss can happen during washes between fractions. That is to be expected, since the washes are done to remove carry over from previous fractions. Following the reviewer's suggestion, we have now analyzed by SDS-PAGE the full extract of each fraction as well as the whole content of each wash. As apparent from the gels below (Panel A below, and Suppl. Fig 4A), it is clear that some protein is lost during the washes, although the contribution of the washes to the total protein amount is almost

negligible. Moreover, we have analysed some of the most prominent bands for each fraction and wash, to validate the assignment of each subcellular compartment, and quantify the proportion of each of these markers that were lost in the washes (Panel B below, and Suppl. Fig. 4B). However, to a certain extent this can be explained by the modestly high positive correlation (Spearman, $R=0.74$) between summed protein intensities across fractions versus protein intensity in whole cell lysate. To make our statement clearer, we have introduced this issue in the text. We have described this in lines 189 to 203 and in Supplementary Figure 4-B.

A

B

Cytosol: EIF3G (O75821) // EKLPGLELPVQATQNK (band 1)

Ribosome: RPL3 (P39023) // HGSLGFLPR (band 1)

Endoplasmic Reticulum: RCN1 (Q15293) // YIFDNVAK (band 1)

Mitochondrion: ATP5F1B (P06576) // VVDLLAPYAK (band 2)

Nucleoplasm: HNRNPA2 (P22626) // GFGDGYNGYGGGPGGGNFGGSPGYGGGR (band 3)

Nucleoli: EBNA1BP2 (Q99848) // GLLKPGLNVLEGPK (band 4)

Line 138-139: Given the continued coupling of fractions 1 with 2, and 3 with 4, it warrants testing whether reducing the fractionation method to 1+2, 3+4, 5 and 6, would suffice. This analysis could be performed in silico on the existing data to determine the feasibility of this.

We appreciate the suggestion of the reviewer, and we agree that in some cases it could be favorable to simplify the subcellular approach merging consecutive fractions. However, we consider that such simplification of the protocol would reduce the resolution significantly, since we would lose information about the separation of nucleoplasm in fraction 5 to nucleolus in fraction 6; or luminal proteins in fraction 3 to membranous proteins in fraction 4. Altogether, we consider that merging consecutive fractions would not imply any benefit for the overall results of the protocol.

Line 168: The authors note that fractionation resolution is highly reproducible between these two cell lines, citing supplementary figure 2E. This statement is largely supported by the data. However, there do appear to be two clusters of proteins that are predominantly in fraction 2 in HeLa cells, but fraction 1 in U2OS. No fractionation procedure is impervious to such deviations, but it may help users of this method to understand where this might come from, to mitigate against it. Could the authors determine if there is a size bias or other property to these protein clusters.

We thank the reviewers for acknowledging the reproducibility of our approach. We have looked further into that cluster of proteins that are purified in fraction 1 in U2OS, but not in HeLa, where they are more abundant in fraction 2 (Panel A-blue cluster). We compared that group of proteins with another group of proteins with similar size that are consistently purified in fraction 1 in both cell lines (Panel A-brown cluster). However, when we plotted the profiles of each group of proteins, we see, that although the distribution of proteins from the blue clusters is more abundant in FR2 in HeLa, they are also present in significant amounts in FR1, and the other way around in U2OS: Which indicates that the heatmap representation is a bit limited, since it does not allow to see clearly this distribution.

Nevertheless, following the reviewer suggestion we explored whether there was any physico-chemical difference in those groups of proteins. Interestingly, we found that proteins that were purified in fraction 2 in HeLa and in fraction 1 in U2OS are significantly bigger in terms of molecular weight (Panel C). Based on that, we hypothesize that cytosolic soluble proteins with higher molecular weight are subject to more variable purification profile depending on the cell line, such as reflected in this case.

The use of MetaMass as a benchmark lends credibility on the one hand, because it was published several years prior to this work, nonetheless, the scoring does appear lenient, which is manifested in two ways. The first is that the F-scores for other datasets are almost all close to 1, the second is that it is clear from plots shown in figure 1F, that membrane bound organelles are not well-resolved, yet the F scores would still imply significant predictive power, as detailed in lines 184-186. The authors should benchmark their method against these methods using an orthogonal metric, for example the QSep function published by the Lilley lab, described in Geladaki et al 2019. In addition, the previous spatial phosphoproteome obtained by protein correlation profiling in the Mann lab should be included in the MetaMass analysis (Krahmer et al 2018), as should work from the Borner lab cited in the introduction.

We thank the reviewer for highlighting the importance of the MetaMass analysis as a benchmark. We have rephrased our statements to indicate better the difference in resolution acquired between other datasets and ours. Also, as mentioned in the answer to the first comment, we have now cited Krahmer et al work (line 73), and compared that dataset using metamass as well (Supplementary Figure 6F).

Moreover, we have used the QSep score described by Gatto and colleagues as an orthogonal approach to calculate the resolution metric in our dataset. QSep score calculation relies on the comparison of the average Euclidean distance of the full, n-dimensional protein profiles within and between subcellular marker clusters. This assumption to calculate the scoring system will therefore benefit those subcellular fractionation methods that generate more fractions than our current approach. Here, we only generate 6 fractions, which of course cannot allow us to clearly resolve the 33 subcellular niches mentioned in the publication. In the original QSep publication, Gatto and colleagues performed an extensive comparative assessment of this score in multiple subcellular proteomics datasets. We employed their approach in our dataset, as well in two datasets that are benchmark in the original paper (Thul et al 2017, Science and Itzhak et al 2017, Cell Reports). In order to avoid potential biases due to the markers used in each case, we re-annotated the markers using those provided by

pRoloc package and calculated the QSep scores for each compartment (Panel A, B and C). When compared to that analysis, our approach obtained a median QSep score of ~2, which is close to the score obtained by the Ithzak et al dataset, but far from the one reported for all the datasets obtained with the hyperLOPIT based on density centrifugation, that scores closer to 5 (Panel D). Based on these results, we can observe that our method shows better capacity to resolve the cytosolic and nuclear compartments, whilst it performs worse when separating the membranous organelles (Panel C). Although the overall scores are worse than in Thul et al (Panel A), in that dataset membrane-organelles also score worse than cytosol and nuclear compartments. This comparison highlights that our current method cannot achieve as high resolution as other subcellular approaches based on differential centrifugation or density gradients, which results in much more fractions. However, it shows that the resolution, measured as QSep score, is on par with other published methods, such as Ithzak et al (2017, Cell reports).

In summary, as stated from the authors of the QSep approach, this metric relies on the definition of as many subcellular niches as possible, which of course is limited in our approach, since we only collect six fractions in order to facilitate overall analysis. Moreover, Qsep authors also point out that their method does not evaluate other important factors in subcellular experiments, such as the possibility to study relative abundance across proteins. Altogether, we consider that QSep is not an appropriate metric to evaluate the performance of our current approach, since the main purpose of our workflow, rather than describe thoroughly and in great resolution the different cellular compartments is to provide a affordable workflow to study events of cellular translocation between the main cellular neighbourhoods in a dynamic manner.

A Thul et al, 2017, Science

B Itzhak et al, 2017, Cell Reports

C Present Study

D

Line 258-274: The authors very elegantly identify a reason that phosphorylated EGFR is observed in the cytosol. However, why the lysosomes are found in the cytosolic fraction, when applying the method to tissues, is not discussed. The difference between the protocol for cultured cells and tissues is the use of the homogenizer, which may disrupt cell integrity. It would be valuable confirm this hypothesis experimentally, or otherwise determine the reason for lysosomes being found in the cytosolic fraction. For example, to monitor cellular integrity, please check the post-homogenization supernatant for cytosolic markers.

We understand the concern of this reviewer, and we have thoroughly evaluated the integrity of cell ultrastructures and organelles after homogenization and each subcellular fractionation treatment. To do so, we have performed transmission electron microscopy in the liver and kidney, as suggested by Reviewer #2 (see comment 23). We took samples after the homogenization steps, but also after the different treatments of the subcellular process. We observed that after homogenization the cellular integrity was disrupted. Therefore, the cell

membrane was broken, exposing the organelles to the different buffers using the sequential extraction. However, even after this disruption we observed some membranous-like vesicles that still contained subcellular components inside. Therefore, these observations explain how some lysosomes are released during the homogenization step leading to their identification in the first fraction, together with cytosolic and soluble proteins.

In panel A and B below there are representative images of Liver and Kidney samples respectively, where it is clearly seen that the cell integrity is lost, but the main organelles remain intact.

We have included this new information in the manuscript (Fig 5C and Suppl. Fig 8B-C) and lines 332-339 and 520-533.

A

B

Minor Revisions

Line 115: 7957 phosphorylation sites were identified from chemically fractionated HeLa cells, which is substantially less than 20,000 sites from unfractionated HeLa cells in the authors' recent publication. One would anticipate that fractionation would serve to increase depth. Please comment on this discrepancy.

To identify 20,000 sites from unfractionated HeLa cells as we recently reported (Bekker-Jensen et al., Nat. Comms. 2020, PMID: 32034161), we combined multiple replicates as well as higher protein input for phospho-enrichment (~200 ug of purified tryptic peptides). Importantly, from a single-shot HeLa the highest coverage at phospho-site level (with confident localization) is of ~6,000 sites (Bekker-Jensen, 2020, MCP, PMID: 32051234).

Although the reviewer makes an interesting point, fractionation does not always provide deeper phospho-proteome profiling. We have observed experimentally that fractionation only increases the depth of a phospho-proteomics analysis when the starting material is higher than that used in a single-shot analysis. In our current approach, the initial starting amount at the phospho-enrichment level per fraction is compromised, not only due to limited initial sample amount, but also due to the losses suffered during the chemical fractionation approach, which hampered the availability of resulting peptides for phospho-enrichment. Therefore, we did not expect to obtain a higher coverage than the one we normally obtain in an unfractionated HeLa using optimal experimental conditions.

Line 115-116, together with lines 620-622: 6952 proteins were quantified in total, yet only 4000 proteins per fraction implies a lot of imputation is required, yet the imputation process is insufficiently described.

We agree with the reviewer, and in fact, imputation is an important part of the data analysis pipeline employed in the current manuscript. We have extended the method section to explain in more detail the imputation process used (Methods section > Data analysis > lines 739-750).

Line 164: the word 'some' is interpreted liberally, it would appear that only three kinase substrate relationships have been analyzed. The authors should conduct a more systematic analysis or remove this statement.

We agree with the reviewer that a more systematic analysis of kinase-substrate relationships are warranted. Consequently, we decided to include a more extended and unbiased analysis of the kinome-substrate subcellular landscape. For that purpose, we annotated all kinases known to phosphorylate substrates identified in our dataset using PSP information with the Perseus computational platform. Then we annotated for each kinase and each substrate phospho-site their main subcellular location and plotted the resulting information as a network (see Panel A below), where kinases are grouped based on their main location (indicated by the color of the outer ring of the node) and their interactions extracted from StringDB. Then we included the substrate information as a pie-chart inside each kinase, where all its known substrates observed in our data are plotted, again shown with a color-code indicating their main subcellular location. We selected some of the most promiscuous kinases in our dataset (those with many known substrates) to visualize in more detail their kinase spatial landscape: GSK3, MAPKAPK2, AURKA, EGFR, PRKAA1. On the one hand, we found that the subcellular distribution of substrates for some kinases are restricted to their main location, such as EGFR

or PRKAA1. However, for other kinases, we show that the location of their substrates does not necessarily match the location of the kinase itself, such as GSK3B or MAPKAPK2. This either suggests the existence of mobility of the kinase to exert its function, or, on the contrary, the mobility of the substrates once they are phosphorylated.

A

B

Line 196: Use of the word comprehensive seems inappropriate, since the depth, although very good for short gradients, still falls short of comprehensive, where data from the Olsen lab serves as the current benchmark.

We have changed the word comprehensive to extensive, to denote that the depth is high but not complete.

Line 218: Translocation of EGFR from the plasma membrane to early endosomes was observed in spatial proteomic studies from the Borner lab, this should be mentioned to highlight the information lost in this approach relative to centrifugation-based approaches.

We have now mentioned the work from the Borner lab on assessing EGF mediated translocation of EGFR using subcellular proteomics (lines 280-281).

Line 219: The authors state that they can 'clearly detect' how adaptor proteins reduce their cytosolic presence, and although the three proteins cited are in the top 2% (80/3883), the BH corrected p-value for SHC1 is higher than the routinely used 5% cutoff. This targeted assessment seems biased by prior knowledge, please adjust wording accordingly, or provide further details as to how hits were triaged.

We agree with the reviewer that the previous version of the manuscript lacked the rationale behind the selection of these markers. Initially, GRB2, SHC1 and CBL were selected based on prior knowledge, since they have been widely described in literature as adaptors that follow EGFR internalization. Moreover, GRB2 and SHC1 were also identified by Itzhak and colleagues as translocating proteins in response to EGF using subcellular proteomics. Since the use of EGF as a signaling model in our manuscript was intended as a validation experiment, we selected those proteins to show that our methodology was capable of recapitulating what was already known. Nevertheless, we have now included an unbiased statistical approach to identify translocating proteins, with which GRB2, SHC1 and CBL proteins arise as confident translocating events, as well as many others.

We have included a description of this approach in the Method section "Identification of translocation events". Briefly, we have combined the statistical significance of the protein level change across time (derived from the limma moderated t-test adjusted p-value for each fraction between time points) with the change of the protein distribution between compartments (termed as "Movement Score" or "MS"). Then we plotted both values, and filtered those proteins with a combined adjusted p-value > 0.05 and a MS > 10%.

First, we calculated the "Movement Score" to rank the proteins according to how much their profiles change between different time points. To do so, we scaled the protein levels at a given time-point to the total abundance. Then we calculated the absolute difference between fractions in each time point against the control/initial condition, which represents the percentage of the protein that changes distribution. We selected the two compartments that showed the highest difference, since those would be the ones between which the protein moves. Then, we classify the potential translocation events in four categories: cytosol-nuclear (blue), cytosol-membrane (green), membrane-nuclear (red) and within the same neighbourhood (gray), to make the visual inspection of the plots easier.

Secondly, based on the fact that proteins move between two compartments, we assumed that we need to consider the change in both places to confidently identify a "moving" protein (i.e., if a protein moves from the cytosol to the membrane, its intensity should decrease in the cytosolic compartment and increase in the membrane one). Therefore, we need to combine the p-value from the two relevant compartments using the Fisher's method, followed by correction for multiple testing by Benjamini-Hochberg. While doing so, proteins such as SHC1 surpass the significance threshold of 5%, since the p-value for cytosolic compartment movement (at 8 minutes) is 3.038932e-04 and the p-value for the membrane-bound compartment (at 8 minutes) is 0.0190969950. Therefore, using Fisher's method, the combined p-value adjusted by BH results in 0.009.

Whilst the movement score evaluates the change in spatial distribution (i.e., translocation between cellular compartments), the combined p-values takes into account the statistical significance of that change. In previous subcellular proteomics papers translocation events have been assessed either by evaluating changes on the protein distribution profile using correlation metrics (Orre et al, Mol Cell, 2019) without further statistical validation, or by complex outlier detection algorithms (Itzhak et al, eLife, 2016). In contrast, here we present a straightforward approach to estimate protein movement that can be easily applied to identify protein changes in time and space. Using this approach, we can see that GRB2, SHC1 and GRB2 appear as translocating proteins at 2 and 8 minutes, and disappear at 20 minutes, matching the expected transient behaviour on EGFR internalization.

#1. Mobility Score

#2. Combined p-value

#3. Translocation Plots

Line 229-233: The authors infer a causal relationship which is not supported by this data alone but based on prior knowledge. EGFR phosphorylation is observed to increase, which is known to provide a docking site for SHC1. SHC1 is observed to increase membrane association, SHC1_Y427 phosphorylation is observed to increase in both the cytosolic and membrane fractions. The conclusion that EGFR phosphorylation of SHC1 is a direct consequence of subcellular translocation cannot be deduced from these observations alone, because the simultaneous increase in the cytosolic pool of SHC1_Y427 could indicate alternative an alternative mechanism.

We agree with the reviewer about the need clarify in the text a potential mechanism for phosphorylation of SHC1 cytosolic form. We have now referred to this in the manuscript (lines 306-309). Whereas we present in the text that the phosphorylation of SHC1 observed in the membrane-enriched fraction might be connected to EGFR internalization due to EGFR stimulation, it is very clear by our data that SHC1 is also phosphorylated in its cytosolic form. This fact might indicate that other tyrosine kinases are active in the cytosol and therefore can phosphorylate the cytosolic SHC1 form. In fact, some cytosolic tyrosine kinases, such as SRC, are known to phosphorylate SHC1 and it has also been shown that SRC can trigger phosphorylation of its substrate upon EGFR activation (Begley et al, 2015, PMID:26551075).

Line 276: replace 'proved' with the word 'demonstrated'

We have corrected it.

Line 346: Unclear where the hypothesis that RNA maturation was responsible comes from. Could the authors please elaborate on this.

We have rewritten the results section that explain the relationship between RNA maturation and accumulation of Large Ribosomal Protein Subunit in the nucleoli in response to osmotic shock.

Line 382: See comment above for line 73-74.

We have now mentioned Krahmer et al work in our manuscript.

Line 437-438: Not clear how morphology and phenotypic differences will affect detergent performance, could the authors clarify this point.

We apologize for the confusion derived from this statement. It was too speculative and we have removed it from the new manuscript version. In order to clarify differences in proteomic distribution of organelles between kidney and liver, we performed transmission electron microscopy at different steps of the subcellular fractionation protocol in tissues (see Reviewer #2 comment 23) and panels A-E for liver and F-J for kidney.

In light of this new data, we can conclude that different tissues, such as liver and kidney, behave differently during the homogenization process. As seen below in panel A and F, corresponding to liver and kidney tissue after homogenization respectively, the liver tissue loses the ultra-cellular integrity completely, exposing the organelles to the action of the different buffers. In contrast, kidney samples, despite losing cellular integrity (none complete plasma membrane can be traced), they maintain certain tissue morphology. In fact, nucleus, mitochondria and other organelles are found contained in higher structures, and therefore protected from the direct action of the buffers used for the sequential extraction. This difference can potentially explain the differences observed, for instance, between mitochondrial profiles in liver and kidney, where liver mitochondrial proteins are found in earlier fractions, and kidney mitochondrial proteins are mostly found after treatment with DDM in fraction 4. We hope this new data would clarify this issue, and we have modified the manuscript accordingly.

Line 588: Why are MS1 scans collected until 1400 m/z if fragmentation is only performed on ions up to 1033 m/z?

We traditionally use MS1 scans that go from 350 to 1400 m/z. However, when implementing DIA methods we observed that most of the proteins were inferred from peptides spanning 361 to 1033 m/z. Therefore, we optimize the window acquisition for DIA to scan through that range. The reviewer is correct, and it is not necessary to keep the m/z range up to 1400 at MS1 level, since peptides bigger than 1033 m/z won't be fragmented and therefore not identified. We will take this into account and correct it in future acquisition methods.

Lines 589-592: Please replace Da with m/z.

This has been corrected.

Line 613: How or why are there protein identifications without valid gene names for the Uniprot database? If this is the case, could the protein names have been used or other identifier.

Some proteins in the Uniprot database do not have associated gene names, due to the status of the curation of that given protein. This means that the protein sequence is identified and the protein existence has been proven somehow, but no gene name is available. Very few proteins do not have a valid gene name. In our dataset those without a valid gene names in our data matrices correspond mainly to contaminants (which we manually discard for further analysis) or to proteins such as E7EVH7 (<https://www.uniprot.org/uniprot/E7EVH7>) for which evidence has been inferred from homology and no gene names have been curated yet. Although, as the reviewer suggested, we can use the protein names or Uniprot identifiers, we cannot use those proteins for biological inference using tools such as GSEA that rely specifically on gene names. That is why, for simplicity while using gene ontology tools we decided to discard the proteins that lack a valid gene name since they only comprise ~0.5% of the total identifications.

REVIEWERS' COMMENTS

Reviewer #1 (Remarks to the Author):

The authors have addressed my concerns satisfactorily through careful detailed explanation and experiments, and I look forward to the publication of their work in Nature Communications.

Reviewer #2 (Remarks to the Author):

The authors responded to most of my concerns and likely to most of the other reviewers' concerns, too.

However, with due respect, how the authors addressed my comment on dual/multiple localized proteins is not how it should be b/c the pure identification of a protein in multiple fractions is not a meaningful indication that a protein actually resides in multiple niches. What should be assessed here is whether a protein is enriched in more than one fraction. There are known examples in literature that could be used to assess whether the presented approach is sensitive enough to reveal their different niches. So the question is whether the resolution and sensitivity of the method allows to determine the subcellular niche of such dual/multiple-localized proteins. I do not expect that the method should allow this but it would add to the description of the pros and cons of the method, which I deem important for the readers.

With this, I think the authors report a valid methodological approach that should be of interest to those trying to understand how intracellular signaling networks are modulated.

Reviewer #3 (Remarks to the Author):

It was an absolute pleasure to read this revised manuscript. The effort the authors have gone to in order to address the comments made by all reviewers is commendable. In several cases, the authors have gone far beyond what I would have expected, and the manuscript certainly feels improved. The kinase network analysis is beautifully presented in supplementary figure 5, the translocation analysis is elegantly simple, the TEM images provide real insights into the chemical fractionation, and the imputation described in the methods is also intriguing. This will be a wonderful contribution to the

field, and one hopes that other scientists will feel compelled to use this accessible method for spatial proteomics.

Point-by-point rebuttal letter

NCOMMS-21-09890-A:

“Spatial-proteomics reveals phospho-signaling dynamics at subcellular resolution” by Martinez-Val et al.

Our answers to the reviewer questions are indicated in blue text.

REVIEWERS' COMMENTS

Reviewer #1 (Remarks to the Author):

The authors have addressed my concerns satisfactorily through careful detailed explanation and experiments, and I look forward to the publication of their work in Nature Communications.

We are happy that Reviewer #1 is satisfied with our revision and glad that she or he recommends publication of our manuscript.

Reviewer #2 (Remarks to the Author):

The authors responded to most of my concerns and likely to most of the other reviewers' concerns, too.

However, with due respect, how the authors addressed my comment on dual/multiple localized proteins is not how it should be b/c the pure identification of a protein in multiple fractions is not a meaningful indication that a protein actually resides in multiple niches. What should be assessed here is whether a protein is enriched in more than one fraction. There are known examples in literature that could be used to assess whether the presented approach is sensitive enough to reveal their different niches. So the question is whether the resolution and sensitivity of the method allows to determine the subcellular niche of such dual/multiple-localized proteins. I do not expect that the method should allow this but it would add to the description of the pros and cons of the method, which I deem important for the readers.

With this, I think the authors report a valid methodological approach that should be of interest to those trying to understand how intracellular signaling networks are modulated.

We appreciate that Reviewer #2 acknowledge that we responded to most of her or his concerns. Moreover, we apologize for not being clear enough when answering her or his concerns about dual/multiple localized proteins.

We agree with Reviewer #2 that pure identification is not a meaningful indication of whether or not a protein actually resides in a given niche. We also agree with the comment that it is protein enrichment in each fraction what would provide information about the localization. In fact, that is indeed the approach we have followed throughout the manuscript in order to show, where proteins reside and how they move. We scaled the intensities of each protein across fraction, and expected that a protein was localized in the fraction, where the protein is most enriched, which corresponds to the fraction where it is most abundant. We have now added a clarification in that regard in the main text (lines 137-139).

Although with certain limitations due to the purification of only six subcellular compartments, our approach can also identify proteins that are known to be present in multiple compartments

simultaneously. In fact, we already showed the dual, but also dynamic, location of EGFR-adaptor proteins SHC1, GRB2 and CBL, which were all found in both the cytosol and the membrane-associated compartment (Figure 4C)

However, to clarify this further, we specifically looked at some proteins known to have dual localization according to the antibody-based fluorescent image analysis described in the publication by Thul et al in Science (“A subcellular map of the human proteome, 2017). As an example of dual/multiple location, that publication refers to proteins CCAR1 and NDUFA9, which are found to localize at the nucleus and the Golgi apparatus or mitochondria, respectively. We extracted the information for those proteins from our datasets, and found that those multiple location matches well. For both CCAR1 and NDUFA9, majority of each protein in FR5 (nucleoplasm) and FR4 (mitochondrion), respectively. Moreover, we can see some contribution of CCAR1 in FR4 (enriched in Golgi proteins), which is especially clear in U2OS cells. Similarly, for NDUFA9, we can see that the compartments with more presence of the protein after FR4 are those corresponding to the nuclear compartment (FR5 and FR6).

Finally, a great example of multiple localization also presented by Thul et al, is the ribosomal protein L19. Thul et al described that the ribosomal protein can be present in both the cytosol, the endoplasmic reticulum and the nucleoli. When we plot the scaled intensity distribution of this protein across our six fractions, we can clearly see that it also in our datasets is distributed across those three subcellular compartments.

These examples show that it is possible to assign multiple protein locations based on the intensity profiles of these proteins across fractions. However, just visual inspection is somehow limited and some further validation of the contribution to each compartment could be necessary, for instance by employing tools such as the one described by Crook et al (Plos Computational Biology, 2018).

We have included this information as a Supplementary Note 1.

Reviewer #3 (Remarks to the Author):

It was an absolute pleasure to read this revised manuscript. The effort the authors have gone to in order to address the comments made by all reviewers is commendable. In several cases, the authors have gone far beyond what I would have expected, and the manuscript certainly feels improved. The kinase network analysis is beautifully presented in supplementary figure 5, the translocation analysis is elegantly simple, the TEM images provide real insights into the chemical fractionation, and the imputation described in the methods is also intriguing. This will be a wonderful contribution to the field, and one hopes that other scientists will feel compelled to use this accessible method for spatial proteomics.

We are really happy about the kind words from reviewer #3 and that she or he appreciate all the effort that we made to address the comments by all reviewers and that the manuscript has improved after this revision.